

# Geometric constraints on tributary fluvial network junction angles

Jon D. Pelletier, Robert G. Hayes, Olivia Hoch, Brendan Fenerty, and Luke A. McGuire

Department of Geosciences, The University of Arizona, 1040 East Fourth Street, Tucson, Arizona 85721–0077, U.S.A.

*Correspondence to*: Jon D. Pelletier (jdpellet@arizona.edu)

**Abstract.** The intersection of two non-parallel planes is a line. Howard (1990), following Horton (1932), proposed that the orientation and slope of a fluvial valley bottom within a tributary network are geometrically constrained by the orientation and slope of the line formed by the intersection of planar approximations to the topography upslope from the tributary junction along the two tributary directions. Previously published analyses of junction-angle data support this geometric model, yet junction angles have also been proposed to be controlled by climate and/or optimality principles (e.g., minimum-

power expenditure). In this paper, we document a test of the Howard (1990) model using ~$10^7$ fluvial network junctions in the conterminous U.S. and a portion of the Loess Plateau, China. Junction angles are consistent with the predictions of the Howard (1990) model when the orientations and slopes are computed using drainage basins rather than in the traditional way using valley-bottom segments near tributary junctions. When computed in the traditional way, junction angles are a function of slope ratios (as the Howard (1990) model predicts), but data deviate from the Howard (1990) model in a manner that we

propose is the result of valley-bottom meandering/tortuosity. We map the mean junction angles computed along valley bottoms within each 2.5 km x 2.5 km pixel of the conterminous U.S.A. and document lower mean junction angles in incised late-Cenozoic alluvial piedmont deposits compared to those of incised bedrock/older deposits. To understand how this finding relates to the geometric model of Howard (1990), we demonstrate that, for an idealized model of an initially unincised landform, i.e., a tilted plane with random microtopography, lower ratios of the mean microtopographic slope to the

large-scale slope/tilt are associated with lower mean junction angles compared to landforms with higher such ratios. Using modern analogs, we demonstrate that late-Cenozoic alluvial piedmonts likely had ratios of mean microtopographic slope to large-scale slope/tilt that were lower (i.e., ~1) prior to tributary drainage network development than the same ratios of bedrock/older deposits (≫1). This finding provides a means of understanding how the geometric model of Howard (1990) results in incised late Cenozoic alluvial piedmont deposits with lower mean tributary fluvial network junction angles, on

average, compared to those of incised bedrock/older deposits. This work demonstrates that the topography of a landscape prior to fluvial incision exerts a key constraint on tributary fluvial network junction angles via a fundamental geometric principle that is independent of any climate- or optimality-based principle.



# 1 Introduction

Many tributary fluvial networks located on alluvial piedmonts of the Basin and Range province of the U.S.A. are parallel or subparallel (Fig. 1). The dashed curve in Figure 1b delineates the bedrock-alluvial contact of the Santa Catalina Mountains near Tucson, Arizona. South and west of this dashed curve, tributary fluvial valleys incised into late-Cenozoic alluvial piedmont deposits of the Santa Catalina Mountains (Dickinson, 1992) are predominantly parallel and sub-parallel. North and east of this curve, tributary fluvial valleys incised into the bedrock of the Santa Catalina Mountains are predominantly dendritic and rectangular. Basins and ranges of this region are separated by normal faults that juxtapose predominantly metamorphic rocks in the ranges with predominantly unconsolidated alluvium near the surface in the piedmonts/basins. In southern Arizona, normal faulting ceased c. 10 Ma (Davis, 1980) and piedmonts have since undergone several cycles of aggradation and incision driven by late-Cenozoic climatic changes and episodic incisions of valley-floor channels that act as the base level for adjacent alluvial piedmont deposits (Bull, 1991; Waters and Haynes, 2001). These cycles have resulted in alluvial piedmont deposits that, immediately post-deposition, were unincised, low-relief landforms sloping gently from the mountain front to the valley-floor channel that have since experienced incision and tributary fluvial network development.

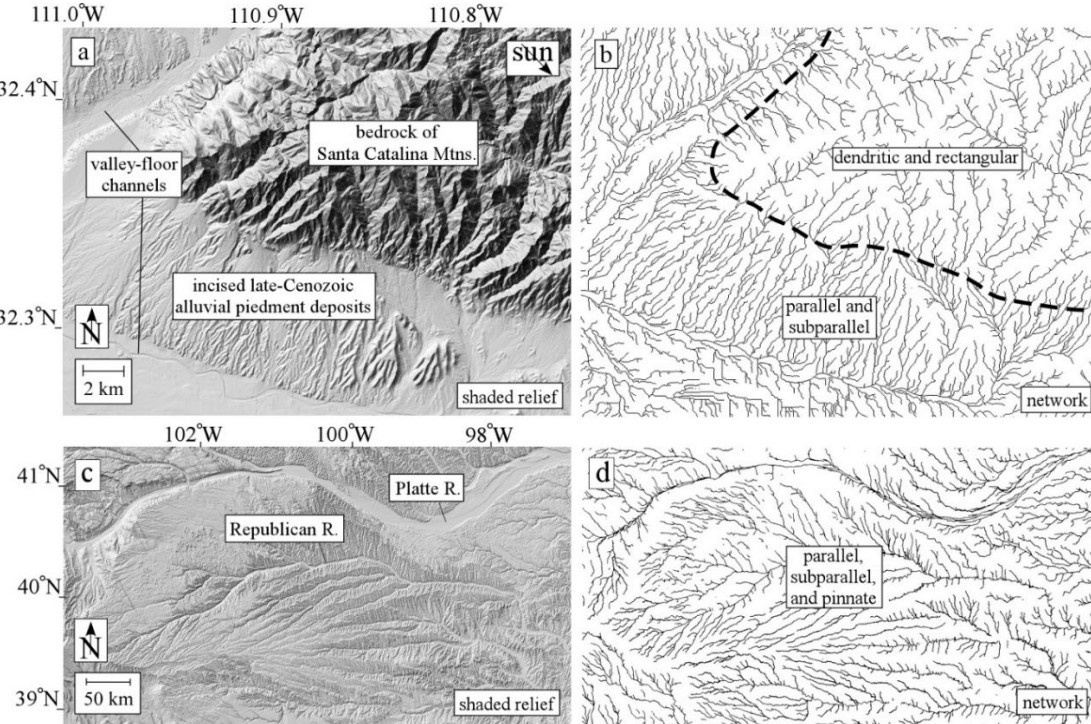

**Figure 1: Shaded relief and fluvial valley network maps of two piedmont regions characterized by predominantly parallel, subparallel, and/or pinnate fluvial networks. (a)&(b) Santa Catalina Mountains and adjacent piedmont comprised of incised late-Cenozoic alluvial deposits; (c)&(d) a portion of the late-Cenozoic alluvial Ogallala Formation in the central Great Plains of southern Nebraska and northern Kansas.**



The piedmont of the Rocky Mountains, i.e., the depozone of the Miocene-to-Pliocene Ogallala Formation of the U.S.A. (Darton, 1899), feature predominantly parallel, subparallel, and pinnate drainage networks (Figs. 1c&1d) (see Zernitz (1932) for a classification of drainage patterns that includes pinnate). As such, both regions illustrated in Figure 1 include incised late-Cenozoic alluvial piedmont deposits with what appear to be relatively low mean tributary fluvial network junction angles.


How might tributary fluvial network junction angles of incised late-Cenozoic alluvial piedmont deposits tend to be lower compared to those of adjacent bedrock/older deposits? In this paper, we test the hypothesis that the lower mean junction angles of late-Cenozoic alluvial piedmont deposits are a consequence of the tendency of their initial, unincised landforms to have lower ratios of mean microtopographic slope to large-scale slope/tilt (Fig. 2). The orientation of a fluvial valley is

initially constrained by the pathways of water flow upslope of the valley, which, for networks dominated by surface runoff, must be a function of the upslope topography. Increasing microtopographic amplitudes, quantified by the root-mean-squared variation in local slope, $S_l$, promotes greater valley tortuosity (Lazarus and Constantine, 2013), which, in turn, may promote larger tributary fluvial network junction angles. Conversely, steeper large-scale slopes/tilts, $S_r$, may promote lower junction angles via the tendency of water flow pathways to be more aligned with the tilt direction as the tilt increases relative to the

mean microtopographic slope that drives local variations in drainage orientations. As such, we hypothesize that $S_l/S_r$ of an initially unincised landform may partly control tributary fluvial network junction angles.

Incised alluvial piedmont deposits are characterized by one or more cycles of aggradation and incision (Bull, 1991). At the end of an aggradational phase, alluvial piedmont deposits tend to be relatively planar, partly as a result of the topographic

diffusion associated with aggradation (Pizzuto, 1987) and the tendency of avulsions to fill in low spots on the piedmont that, according to the control of junction angles by $S_l/S_r$ tested here, may be associated with more subparallel-to-parallel surface-water-flow pathways. Quantitatively, the relief of alluvial piedmonts undergoing active transport and deposition over geologic time scales (i.e., those with predominantly Holocene deposits) is dominated by bar-and-swale topography with amplitudes of ~1 m over spatial scales of ~100 m (Frankel and Dolan, 2007)) while large-scale slopes/tilts are typically on

the order of one to several percent. As such, if alluvial piedmonts with Holocene deposits are adequate modern analogs for the initially unincised late-Cenozoic alluvial piedmonts that have since experienced base-level drop and tributary fluvial drainage network development, the initial $S_l/S_r$ values for late-Cenozoic alluvial piedmonts are likely to be less than or equal to ~1. Bedrock landforms, in contrast, are generally influenced by complex patterns of faulting and folding that often preclude any substantial degree of large-scale planarity. That is, $S_l/S_r$ is likely ≫1 at all stages of the development of fluvial

valleys incised into bedrock/older deposits.



Castelltort et al. (2008) and Castelltort and Yamato (2013) demonstrated the importance of $S_l/S_r$ on the length-to-width ratio of drainage basins using digital topographic analysis and numerical modeling. In this paper we test the applicability of this concept to tributary fluvial network junction angles.


Seybold et al. (2017) and Hooshyar et al. (2017) documented mean tributary fluvial network junction angles between approximately 45° and 72° (in Seybold et al., 2017) and 49.5˚ and 75.0˚ (in Hooshyar et al., 2017). Seybold et al. (2017; 2018) attributed the variation between 45° and 72° primarily to climate (with lower mean junction angles in more arid regions). Hooshyar et al. (2017) attributed the variation in mean junction angles to process dominance (with lower mean

junction angles in areas where incision is driven predominantly by debris flows). Getraer and Maloof (2021) demonstrated that a higher correlation exists between mean junction angles and the ratio of the slopes of the main and tributary valleys than between mean junction angles and the aridity index (defined as the ratio of mean annual precipitation to potential evapotranspiration, such that higher values of the aridity index are less arid), underscoring the likely importance of upslope topography on tributary fluvial network junction angles. Li et al. (2023) argued that tectonic tilting can overprint the role of

climate in controlling junction angles on the steep margin of the eastern Tibetan Plateau. Further clarifying and quantifying the roles of initial topography, climate, and tectonic forcing in controlling junction angles is necessary to better understand this fundamental aspect of fluvial topography and to improve our ability to assess the extent to which junction angles may record information about climate and/or tectonics.

We begin by reviewing the geometric model for junction angles proposed by Howard (1990), following Horton (1932). This model provides a basis for quantifying how upslope topography, including the $S_l/S_r$ of the initially unincised landform, may partly control tributary fluvial network junction angles. We then propose a novel drainage network extraction algorithm that enables the construction of a dataset of ~$10^7$ junction angles for the conterminous U.S.A. We document the importance of the presence/absence of incised late-Cenozoic alluvial piedmont deposits on junction angles, using southern Arizona and the

conterminous U.S.A. as examples. We also consider whether late-Cenozoic aeolian deposits exhibit junction angles similar to those of late-Cenozoic alluvial piedmont deposits, using a portion of the Loess Plateau, China as an example. We then systematically evaluate the relationship between mean junction angles and $S_l/S_r$ on tilted planar surfaces with random microtopography to test whether the signatures of the geometric model of Howard (1990) are present even in the drainage pathways that exist before any fluvial incision takes place.




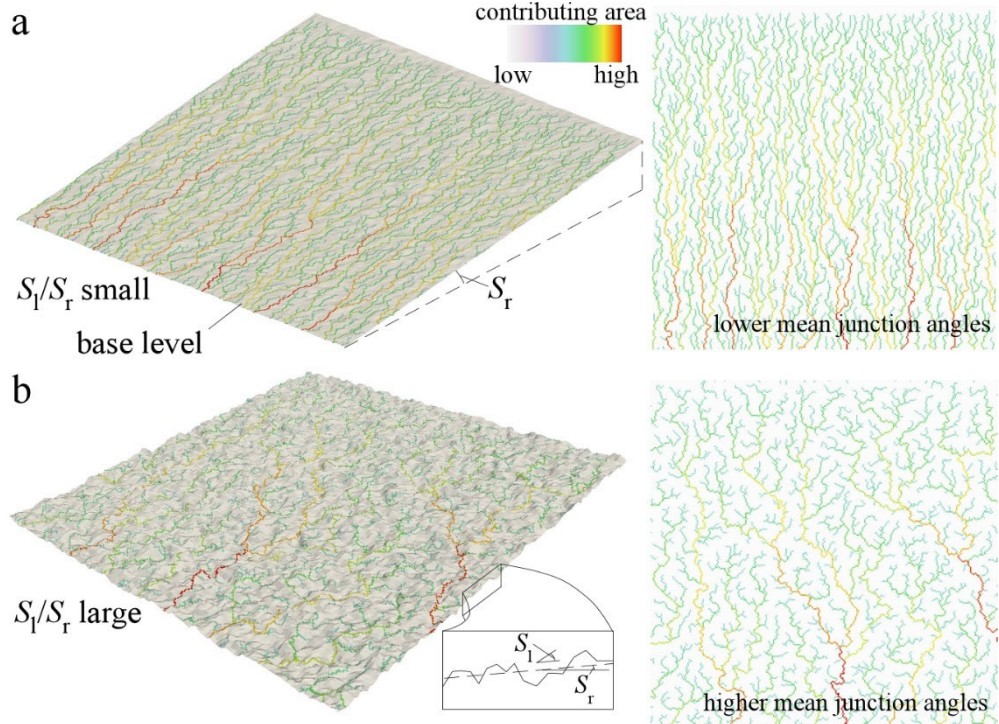

**Figure 2: Conceptual illustration of how differences in the ratio of microtopographic amplitude (quantified by the root-mean-squared variation in local slope, $S_l$) to the large-scale or regional slope, $S_r$, may control fluvial network junction angles. The landforms illustrated in both (a) and (b) have small-scale random microtopography superimposed on a planar tilted slope. The flow patterns defined by contributing area were determined by hydrologic correction and the steepest-descent routing algorithm. Landforms with a lower $S_l/S_r$ (shown in a) result in more parallel fluvial valleys compared to landforms with a higher value of $S_l/S_r$ (shown in b). The specific examples in this figure are ours but the concept closely follows Castelltort and Yamato (2013).**

## 2 Methods

### 2.1 The modified-geometric model and junction-angle extraction from Digital Elevation Models (DEMs)

#### 2.1.1. The modified-geometric model (MGM) for junction angles

Horton (1932) proposed that the junction angle between a tributary valley bottom and a main valley bottom is determined by the intersection of the paths of steepest descent of planar approximations to the topography upslope from each tributary junction. Horton's geometric model was limited in that it assumed that the main valley had the same orientation upstream and downstream of the tributary. Howard (1990) rectified this limitation by modifying the model of Horton (1932) to include two tributaries joining together to make a larger main valley with an orientation downstream of the tributary junction that is distinct from that of either of the two valleys upstream from the tributary junction. In this paper we refer to Howard's modification of Horton's geometric model as the modified-geometric model (MGM).



In the MGM, the orientation and slope of a main valley bottom is defined by the intersection of two planes, each an approximation to the topography upslope of the tributary junction along the two directions of largest upslope contributing area. In this paper we test two versions of the MGM: one in which the topography upslope along each of the tributary directions is the entire drainage basin (denoted as BA for basin-averaged) and another (i.e., the traditional approach) in which the topography upslope along each of the tributary directions is limited to valley bottom segments in the vicinity of

the tributary junction (denoted as AVB for along-valley bottom) (Fig. 3).

The vector defining the intersection of any two planes is the cross product of the normal vectors of the planes. Howard (1990) demonstrated that the MGM predicts that the cosine of each tributary junction angle is equal to the ratio of the slopes of the main (labeled as 3) and tributary (labeled as 1 and 2) valley bottoms (Fig. 3):

$$\theta_1 \approx \mathrm{acos}\left(\frac{S_3}{S_1}\right), \theta_2 \approx \mathrm{acos}\left(\frac{S_3}{S_2}\right) \tag{1}$$

Equation (1) states that, as the slope between the tributary and main valley become more similar, so must their planform orientations. This is not a trivial or obvious relationship, in part because the slope is a function solely of steepness and orientation is a function solely of planform characteristics (i.e., it does not depend on any vertical aspect of the landform). The approximate signs in equation (1) reflect the fact that equation (1) is an approximation to the cross product of the normal

vectors of the planes. This approximation is nearly exact for all slopes that are smaller than $\approx 60°$ (i.e., essentially all fluvial valleys).

Recent analyses of junction angles (e.g., Seybold et al., 2017; 2018; Hooshyar et al., 2017; Getraer and Maloof, 2021) have considered the sum of the two tributary junction angles defined by Howard (1990), i.e., $\theta_1 + \theta_2$. Measuring $\theta_1$ and $\theta_2$

separately provides more complete information about the geometry of the junction (i.e., $\theta_1 + \theta_2$ quantifies how the two tributary orientations relate to one another but not how either tributary orientation relates to the main valley orientation downstream of the junction) and is necessary for testing the MGM.

The blue curves in Figure 3 illustrate the AVB flow pathways along each of the three directions emanating from the tributary

junction. Thin white lines illustrate how the orientations, $\theta$, and slopes, $S$, along the three directions are calculated as linear approximations to what may be tortuous AVB flow pathways. Salmon-colored shaded areas in Figure 3 illustrate the two drainage basins upslope from the tributary junction that are used to compute BA properties along the upslope directions 1 and 2. The BA properties defined along direction 3 are computed using the total area of drainage basins 1 and 2.





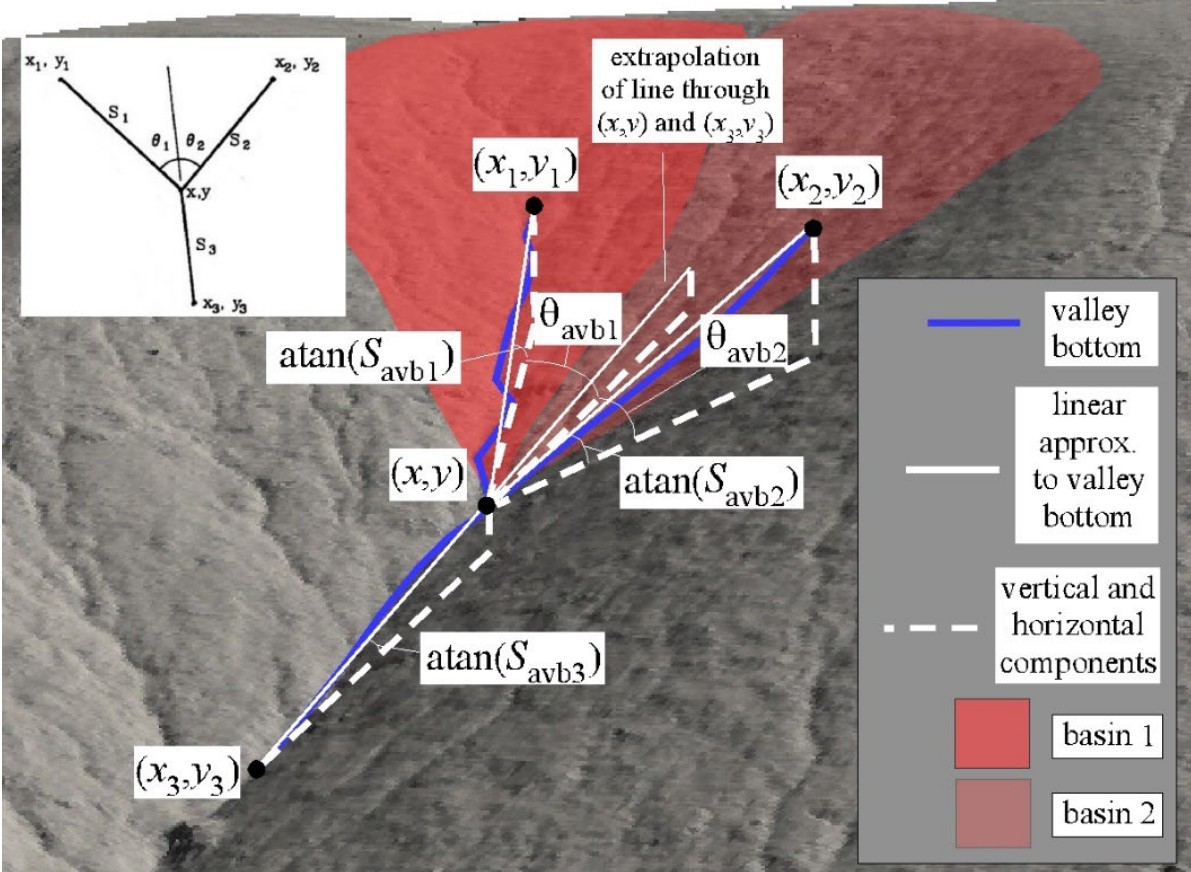


**Figure 3: Schematic diagram illustrating the AVB junction angles θ$_{avb1}$ and θ$_{avb2}$ and their relationships to the steepest-descent pathways in the direction downslope from the tributary junction (labeled 3) and the directions upslope in the direction of the largest contributing area (direction 1) and the second largest contributing area (direction 2). Also illustrated are the basins 1 and 2 used to compute BA junction angles θ$_{ba1}$ and θ$_{ba2}$. Inset diagram illustrating the AVB junction angles in map-view is from Howard**
**(1990).**

*2.1.2. Drainage network extraction and junction angle measurement*

We developed a novel algorithm for junction-angle extraction from a DEM. This algorithm identifies tributary junctions in four steps. First, all areas of internal drainage that are less than a threshold maximum depth (10 m is used here) are assumed
to be areas that are noise/errors in the DEM, areas of anthropogenic infrastructure/disturbance, etc., that are best treated by hydrologic correction (the recursive fill-and-spill procedure of Pelletier (2008) is used here). Areas of internal drainage with depths larger than the prescribed threshold maximum value are assumed to be true depressions and are not filled, resulting in disconnections in the fluvial network at the downstream spill points of those areas of internal drainage. Second, contributing areas are computed for each pixel in the DEM using steepest-descent flow routing. Third, a user-prescribed threshold
contributing area (0.1 km$^2$ is used here but the sensitivity of the results to this value was determined by repeating the analyses with 0.3 km$^2$) is used to define valley heads. Fourth, for each valley bottom pixel (*i,j*) downslope from each valley





head, we compute the ratio of the sum of the two largest contributing areas of the nearest neighbors (including diagonals) to the contributing area of pixel (*i,j*). If this ratio is larger than or equal to a specified threshold (0.99 is used here), the pixel is treated as a tributary junction. The ratio 0.99 means that the total contributing area from pixels other than the two largest

tributaries is less than 1% of the total contributing area in pixel (*i,j*).

For every tributary junction thus defined, the algorithm identifies the direction of steepest descent (direction 3 in Fig. 3) and the directions of the largest (direction 1) and second-largest (direction 2) contributing areas among the nearest-neighbor pixels upslope. To compute the AVB junction angles and slopes, the algorithm searches along each of the three steepest-

descent pathways (one downslope and two upslope) until the elevation change between the tributary junction and the location along each search direction is larger than a threshold value (10 m is used here but the sensitivity of the results to this value was evaluated by repeating the analyses with 5 m and 30 m). The default value of 10 m of elevation change was chosen to be sufficiently small that local orientations and slopes are being calculated but large enough that the method is not substantially biased by elevation errors/noise in the DEM. When computing BA properties, the algorithm computes the

average orientation and slope of the entire drainage basin whose outlet is that junction, using every pixel upslope along directions 1, 2, and 3 (the latter being the total area comprising drainage basins 1 and 2).

## 2.2 Analyses performed on natural landforms

The analyses of this paper include both natural and synthetic landforms. The natural landforms include: Holocene alluvial

piedmonts of the Ft. Irwin region of California, a portion of the Basin and Range Province of southern Arizona, a portion of the Loess Plateau in China, and the conterminous U.S.A. (CONUS).

### 2.2.1 Holocene alluvial piedmonts of the Ft. Irwin region

Random variations in initial topography, in addition to spatial variations in erodibility and tectonic forcing, result in

tortuosity in fluvial valley bottoms that we hypothesize partly control junction angles. To investigate the potential impact of the microtopography of the initially unincised landform on fluvial network junction angles using numerical modeling, we must quantify the statistical nature of that microtopography so that we can create synthetic realizations for hypothesis testing.

We posit that Holocene alluvial piedmont deposits are an appropriate analog for the initially unincised state of late-Cenozoic

alluvial piedmont deposits that have experienced tributary fluvial network development. Areas of Holocene deposits include active channels and adjacent areas that may be flood-prone during extreme flow events. They are are distributary in nature, while Plio-Pleistoocene deposits are typically tributary in nature due to climate-change-driven base-level changes associated with valley-floor-channel incision downstream and the fact that sufficient time has elapsed for tributary fluvial network development to occur on these deposits (Christensen and Purcell, 1985).




In this section, we quantify the microtopography of Holocene alluvial piedmonts of the Ft. Irwin region of California because the piedmont deposits of that area are nearly all Holocene in age (Miller et al., 2013). In contrast, alluvial piedmont deposits in other portions of the Basin and Range province of California tend to be predominantly Plio-Pleistocene in age (e.g., Death Valley; Workman et al., 2002). We focused on the Basin and Range Province in California for this analysis

because surficial geologic maps that distinguish Holocene and Plio-Pleistocene deposits tend to be more widely available for this region compared to other parts of the Basin and Range.

The simplest model of microtopography is one in which the elevation of adjacent pixels is uncorrelated (i.e., white noise). White-noise microtopography is not a realistic model for the microtopography of natural landforms, however, because

spectral analyses of natural landforms demonstrate a generally inverse relationship between power-spectral amplitude and wavenumber (e.g., García-Serrana et al., 2018, Luo et al., 2021). In this study, we performed power-spectral analyses of along-strike transects of the microtopography of Holocene surfaces of the Ft. Irwin region using a 1 m pixel$^{-1}$ DEM derived from airborne-lidar data obtained from the natural resources staff of Ft. Irwin. The power-spectral behavior of microtopography thus constrained, we generated synthetic microtopography with statistical properties identical those of the

Holocene alluvial piedmonts of the Ft. Irwin region for use in the junction angle analyses of tilted planar landscapes with microtopography (Section 2.3.2).

### 2.2.2 Southern Arizona

The motivating example in Figures 1a&1b suggests that junction angles may be systematically lower, on average, in fluvial

networks incised into late-Cenozoic alluvial piedmont deposits than those incised into adjacent areas of bedrock/older deposits. We analyzed a portion of southern Arizona that includes several mountain ranges and their intervening piedmonts/basins to determine whether the lower mean junction angles of piedmonts comprised of incised late-Cenozoic alluvial deposits suggested by Figures 1a&1b can be confirmed quantitatively and over a larger region. We used the data from the National Elevation Dataset (Gesch et al., 2002) for this purpose, projected to a UTM coordinate system at 30 m

pixel$^{-1}$ resolution.

### 2.2.3 Loess Plateau

We included an analysis of the tributary fluvial network junction angles of a portion of the Loess Plateau, China, in this study for two reasons. First, this region allows us to test the MGM in fluvial networks incised into an unusually

homogeneous substrate (i.e., a well-sorted silt-sand deposit). Second, as an aeolian deposit, results from the Loess Plateau enable us to test whether the MGM is applicable to fluvial network development into both aeolian and fluvial deposits. We used 90 m pixel$^{-1}$ DEM data from the Shuttle Radar Topographic Mission (Farr et al., 2007) for this purpose.



Landform evolution in the Loess Plateau is characterized by a competition between fluvial erosion and aeolian deposition
from approximately 3 Ma to the present. The Loess Plateau was a low-relief bedrock landform c. 3 Ma (Xiong et al., 2014)
when climatic changes associated with the development of Northern Hemispheric ice sheets increased the rate of dust
deposition (Nie et al., 2015). Since then, fluvial valleys in the Loess Plateau region with relatively large contributing areas
have been able to keep pace with aeolian deposition (large rivers such as the Ji follow the contact between the loess and the
underlying Cretaceous bedrock closely, see Section 3.1.4) while hillslopes and fluvial valleys with relatively small
contributing areas have not kept pace with aeolian deposition, resulting in loess aggradation.

*2.2.4 Conterminous U.S.A. (CONUS)*

The input DEM for junction-angle extraction for CONUS was created by downloading and merging individual tiles from the
National Elevation Dataset (Gesch et al., 2002). We projected the merged DEM to the Lambert Conformal Conic (LCC)
projection at 50 m pixel$^{-1}$ resolution. The LCC projection was chosen because it is optimally angle-preserving for large
regions (Seybold et al., 2017; 2018).

**2.3 Synthetic landforms**

*2.3.1 Idealized branching network landform*

We validated the drainage-network-extraction algorithm of this paper on an idealized branching network with known
junction angles. The idealized branching network used for this purpose was constructed by first digitally drawing a tributary
network of known junction angles using the graphics program Canvas. That digital image file, with valley-bottom pixels
assigned a value of 1 and non-valley-bottom pixels assigned a value of 0, was then used as input to a simple landform
evolution model built from components described in Pelletier (2008) that include topographic diffusion and a uniform and
constant vertical uplift rate in all non-valley-bottom pixels (Fig. 4).



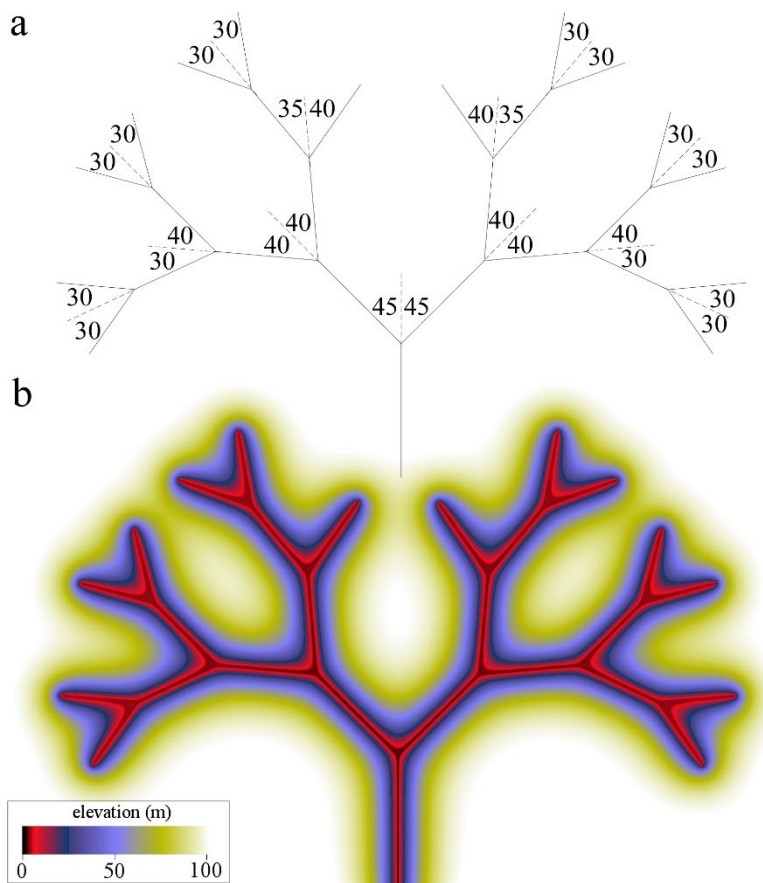

**Figure 4: Images of the idealized branching network used to test the junction-angle-extraction algorithm. (a) network, illustrating examples of the three types of junction angles present in the network. (b) Color map of the topography of the synthetic landform.**


### 2.3.2 Planar tilted landforms with random microtopography

The second type of synthetic landform consider in this paper is random microtopography of a prescribed Gaussian distribution with root-mean-squared variation in local slope, $S_l$, superimposed on a plane tilted to a prescribed large-scale

slope/tilt $S_r$. Hydrologic correction is performed on these and all other landforms analyzed in this paper, with the difference in the case of these synthetic landforms being that all depressions of any size are filled in. The junction angles of the steepest-descent pathways of such tilted planar landscapes with microtopography and hydrologic correction are instructive to consider because they have not experienced any geomorphic evolution, hence any drainage patterns they exhibit can be associated with fundamental, non-geomorphic principles.


We used the Fourier-filtering method (e.g., Malamud and Turcotte, 1999) to generate microtopography that matches the observed power-spectral form of Holocene alluvial piedmonts documented in Section 3.1.1. This method uses a pseudo-





random-number generator to produce white-noise microtopography with a Gaussian distribution of values, transforms the data into wavenumber space using a 2D Fast Fourier Transform, multiplies each Fourier coefficient by the square root of the
wavenumber-dependent square root of the power spectrum, and then inverse-transforms the data back to real space.

## 3 Results

### 3.1 Natural landforms

*3.1.1. Power-spectral analysis of Holocene alluvial piedmonts of the Ft. Irwin region of California*

Figure 5 plots the average power-spectral density of the along-strike topographic variations of two Holocene alluvial
piedmonts in the Ft. Irwin region of eastern California. The Holocene age and alluvial nature of these areas is based on surficial geologic mapping by Miller et al. (2013).

Figure 5b plots the power-spectral density, $S_p$, averaged across all topographic transects along the N-S direction, as a function of natural wavenumber, $\nu$, for spatial scales of approximately 1-1000 m. The power spectra in both cases are similar
to a Brownian walk, i.e., $S_p(\nu) \propto \nu^{-2}$, with the possible exception of a transition to a constant power-spectral density at the largest spatial scales (i.e., smallest wavenumbers). We allowed for the possibility of such a transition when generating synthetic microtopography by adopting the power-spectral model (termed a Lorentzian function):

$$S_p \propto (\nu^2 + \nu_0^2)^{-1} \qquad (2)$$

where $\nu_0$ is the wavenumber of the transition from constant to Brownian power-spectral behavior and low and high
wavenumbers, respectively. We chose to include this transition, despite limited evidence for it in the data of Figure 5, because Brownian walk variability tends to result from avulsions and the along-strike topographic diffusion characteristic of alluvial sedimentary basins (e.g., Pelletier and Turcotte, 1997), but only up to spatial scales associated with the spacing between adjacent drainage basins that source the piedmont or sedimentary basin. Above that spatial scale, the result of fluvial deposition is a bajada or series of coalescing alluvial fans with along-strike topography that can be expected to have
reduced variance relative to a Brownian walk at the largest spatial scales. We generated synthetic microtopography with a Lorenztian power-spectrum and a prescribed root-mean-squared variation in local slope, $S_l$. These synthetic microtopographic examples were each superimposed on planes with a prescribed tilt, $S_r$.





**Figure 5: Quantification of the power-spectral properties of example Holocene alluvial piedmonts in the Ft. Irwin region of eastern California. (a) Shaded relief image of example areas with zoom-in on transects shown on graphs. (b) Elevation, $z$, versus distance, $x$, of example transects. c) Plot of the power spectrum, $S_\mathrm{p}$, as a function of the natural wavenumber, $\nu$, for the landforms in (a). Also plotted are the power spectra associated with a Brownian walk and a Lorenztian, i.e., a Brownian walk that transitions to a constant spectrum at low wavenumbers.**

### 3.1.1 Comparison of southern Arizona valley networks with NHDPlusV2

Figure 6 compares the valley networks resulting from the junction-extraction algorithm of this paper to that of the NHDPlusV2 dataset for the larger Tucson region. Figure 6 illustrates that the algorithm of this paper extracts many more junctions than are mapped in NHDPlusV2. In addition, the algorithm results in a more uniform coverage of valleys compared to NHDPlusV2, which has spatial variations in drainage density that do not correspond to actual variations in drainage density readily identified in shaded-relief images (Fig. 6a). A comparison of Figures 6b&6d indicates that the



results of the junction extraction algorithm are not sensitive to the resolution of the input DEM data between resolutions of 30 and 50 m pixel$^{-1}$.

We used a threshold contributing area of 0.1 km$^2$ to identify valley heads because it results in fluvial valleys in the
Tucson region that are similar to those that we would have identified by visual inspection. To determine whether the results are sensitive to this threshold, we repeated our analyses with an alternative value of the threshold area equal to 0.3 km$^2$.

**Figure 6: Comparison of the tributary fluvial valley bottom networks for the larger Tucson region. (a) Shaded-relief image of the**
**30 m pixel$^{-1}$ National Elevation Dataset (NED). Fluvial valley networks obtained in this study using (b) 30 m pixel$^{-1}$ NED data, (c) the NHDPlusV2 data (McKay et al., 2012), and (d) using 50 m pixel$^{-1}$ DEM data.**



*3.1.2 Dependence of mean junction angle on the presence/absence of Plio-Quaternary alluvial piedmont deposits in*
*southern Arizona*

Figure 7 illustrates the results of the junction-angle-extraction algorithm for a portion of southern Arizona. A visual comparison of the map of the geometric mean of all junction angles within each 2.5 m x 2.5 km square (Fig. 7b) to that of the presence/absence of Plio-Quaternary alluvial piedmont deposits indicates that mean junction angles are typically in the range of 15˚–25˚ (red and dark blue in the color map of Figure 7b) in Plio-Quaternary alluvial piedmont deposits of southern
Arizona, while mean junction angles in networks incised into bedrock/older deposits are in the range of 35˚–45˚ (medium-to-light blue in Fig. 7b). Note that we are using the term junction angle to refer to angles $\theta_1$ and $\theta_2$ individually to be consistent with Howard (1990), not $\theta_1 + \theta_2$ as other recent studies have done. We use the term Plio-Quaternary to refer to the range of ages of piedmont deposits in southern Arizona and late-Cenozoic to refer to the range of age of pediment deposits in CONUS because piedmont deposits in southern Arizona are almost all Plio-Quaternary in age, while CONUS include large
deposits of Miocene age, including the vast Ogallala Formation of the Great Plains. The highest mean junction angles in southern Arizona are in the range of 60˚-90˚ (yellow to white in the color map of Fig. 7b) and are associated with valley-floor channels where two adjacent piedmonts of opposing orientations intersect; these special cases will be further discussed in Section 3.1.3. Figure 7d plots the aridity index from Trabucco and Zomer (2019). A Spearman correlation analysis (Spearman, 1904) demonstrates that the mean junction angle computed at the 2.5-km scale is more strongly correlated with
the presence/absence of Plio-Quaternary alluvial piedmont deposits (Spearman correlation coefficient of $\rho = 0.12$ and $p$ value of $\sim 10^{-43}$) than with the aridity index ($\rho = 0.04$ and $p = \sim 10^{-5}$). The presence of Plio-Quaternary alluvial piedmont deposits was assigned a value of 0 and the absence of Plio-Quaternary alluvial piedmont deposits was assigned a value of 1 for this analysis, hence the positive value of $\rho$ is associated with a lower mean junction angle for fluvial networks incised into Plio-Quaternary deposits than for those incised into bedrock/older deposits. Essentially identical results were obtained
analysis when the analysis was repeated on a fluvial valley network extracted using a threshold contributing area of 0.3 km², i.e, the Spearman correlation coefficient is $\rho = 0.11$ and the map of mean junction angles is visually indistinguishable from Figure 7b.

It is important to emphasize that the presence/absence of late-Cenozoic alluvial piedmont deposits is a proxy for what we
hypothesize is the primary control on junction angles: initial $S_l/S_r$. Lower initial $S_l/S_r$ values are likely associated with late-Cenozoic alluvial piedmont deposits compared to bedrock/older deposits because such landforms tend to have a relatively low microtopographic amplitude prior to incision as a result of the avulsions and topographic diffusion associated with aggradation, e.g., local variations in elevation of ~1 m over spatial scales of ~100 m, as discussed conceptually in Section 1 and documented in the example data of Section 3.1.1.





**Figure 7: Mean along-valley-bottom (AVB) junction angles and potential controlling variables for a portion of the Basin and Range province in southern Arizona. (a) Shaded relief image. (b) Color map of mean junction angles obtained by averaging the angles of all junctions in each 1 km² subdomain. (c) Map illustrating Plio-Quaternary alluvium and bedrock/older deposits. (d) Color map of the aridity index.**

*3.1.3 Comparison of southern Arizona valley networks to the predictions of the modified-geometric model*

Figures 8a&8b plot junction angles as a function of slope ratios for southern Arizona. We use a logarithmic scale for the y axis of Figure 8a not to suggest any particular functional form of trends in the data but merely to spread out the data points that would otherwise cluster in the lower right corner of the graph and therefore be difficult to distinguish. Figures 8a&8b illustrate a generally inverse relationship between junction angles and slope ratios, i.e., when a relatively steep tributary joins with a main valley of much lower slope, the along-valley junction angle tends to be close to 90°. Conversely, when the incoming and outgoing valley bottom to a tributary junction have similar slopes, the junction angle approaches zero. There is





substantial scatter in the data. This scatter could reflect the imperfect nature of planar approximations to drainage basins, the local meandering/tortuosity of valley bottoms, geological heterogeneities that influence landform orientations over a range of spatial scales, etc.


Figure 8c plots the mean junction angles for AVB and BA properties, averaged in bins of slope ratio (each is 0.033 wide for a total of 30 bins from a slope ratio of 0 to 1). The plot of mean BA junction angles closely follows the prediction of the MGM (eqn. (1)). This result indicates that when the two upslope tributary drainage basins are approximated as planes, the intersection of those planes defines the slope and orientation of the drainage basin formed by the union of the two tributary

drainage basins. The mean junction angle calculated using AVB properties is systematically shifted to the left relative to the curve for BA properties, i.e., for the same value of the slope ratio, AVB junction angles tend to be similar for the end-member cases of slope ratios close to 0 and 1 but are lower than the BA junction angles for cases in which the slope ratios are mid-ranged, i.e., 0.4–0.6. In Section 3.2.5 we will delve more deeply into the possible reasons for this shift and the dependence of the results on the elevation change over which the AVB junction-angle data are computed.


The presence of relatively large mean junction angles along large valley-floor channels in southern Arizona such as the Santa Cruz and San Pedro Rivers (locations in Fig. 7b) is an exception to the tendency of Plio-Quaternary alluvial piedmont deposits in southern Arizona to have lower junction angles. This exception is, however, consistent with the MGM because adjacent valley bottoms within a single piedmont tend to have slopes similar to each other and to the large-scale slope of the

piedmont (typically on the order of $10^{-2}$ m m$^{-1}$), but when the relatively steep piedmont valleys join with large valley-floor channels such as the Santa Cruz and San Pedro Rivers (which have slopes $\sim 10^{-4}$ to $10^{-3}$ m m$^{-1}$), the slope ratios $S_{avb3}/S_{abv1}$ and $S_{avb3}/S_{abv2}$ will typically be $\sim$0.01-0.1. The MGM accurately predicts mean junction angles of close to 90° for such junctions involving valley-floor channels.






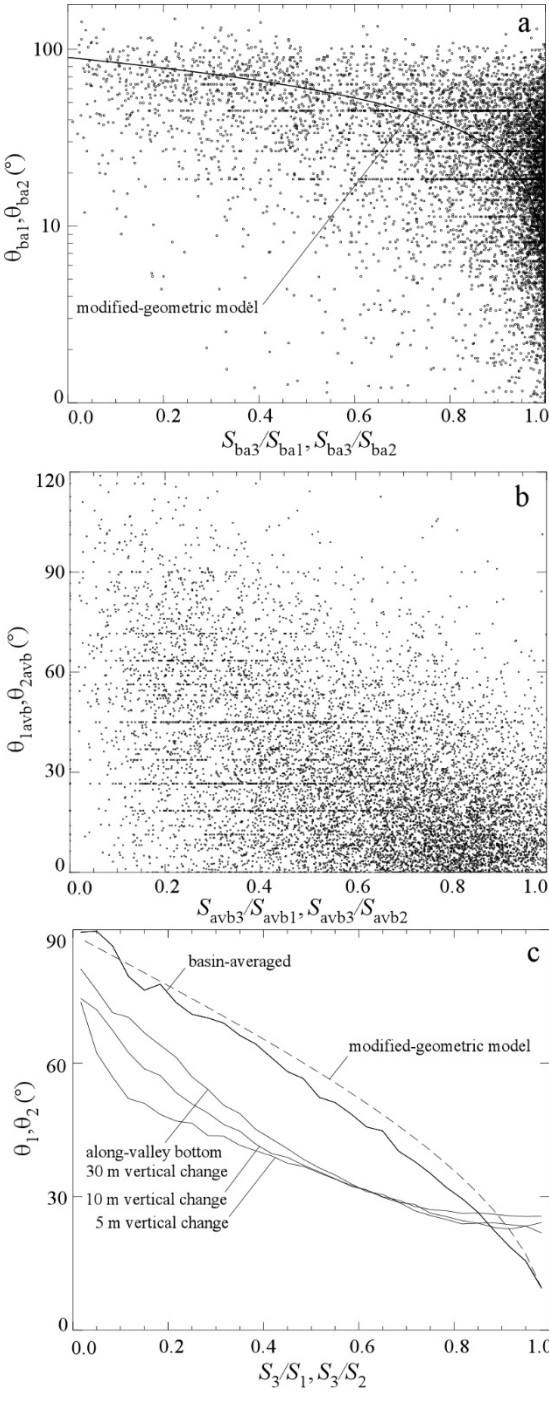

**Figure 8: Plots illustrating the relationships of junction angles to the ratios of slopes downslope and upslope of the junctions for the portion of southern Arizona illustrated in Figure 5. (a) Plot of junction angles measured using BA properties as a function of the ratio of slopes downslope and upslope. (b) Plot of junction angles measured using AVB properties as a function of the ratio of slopes downslope and upslope. (c) Plots of junction angles measured both AVB (using three different values of the elevation change over which slopes and orientations are computed) and BA properties.**




### *3.1.4 Results for a portion of the Loess Plateau, China*


Figure 9 illustrates the results of the junction-extraction algorithm for a portion of the Loess Plateau, China. Figure 9d illustrates the same types of plots for the Loess Plateau as were presented in Figure 8b for the southern Arizona region. The results are essentially identical, i.e., the relationship between the mean BA junction angles and slope ratios follow the MGM

closely while the AVB data are shifted to the left and have a concave-up rather than a concave-down relationship between junction angle and slope ratio.

Another way of quantifying the dominant role of slope ratio is to plot probability density functions of AVB junction angles for several different ranges of slope ratios (Fig. 9c). For slope ratios less than 0.1, AVB junction angles have a peak in the

distribution of values of approximately 80˚-90˚. For increasing slope ratios, the peaks in the distributions of junction angles systematically decline to lower values. The distributions obtained for portions of CONUS (not shown) are less systematic than those plotted in Figure 9d, consistent with the hypothesis that the relatively straight (low tortuosity) valley bottoms of the Loess Plateau region result in an unusually close correspondence between junction angles and slope ratios. The similarity between results from the Loess Plateau and southern Arizona suggest that the trends we observe are not specific to a

particular geographic area or to the processes responsible for the deposition of the substrate (e.g., aeolian versus fluvial) into which fluvial-network development has occurred.



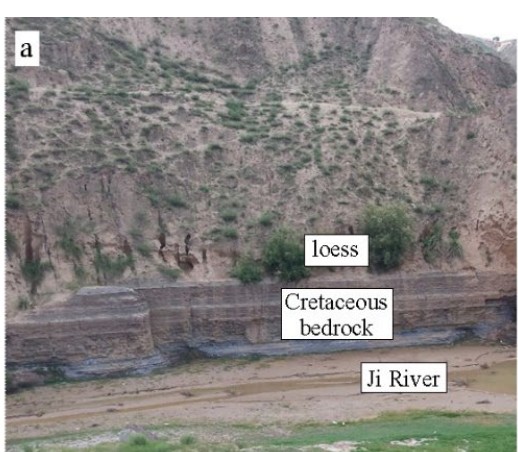

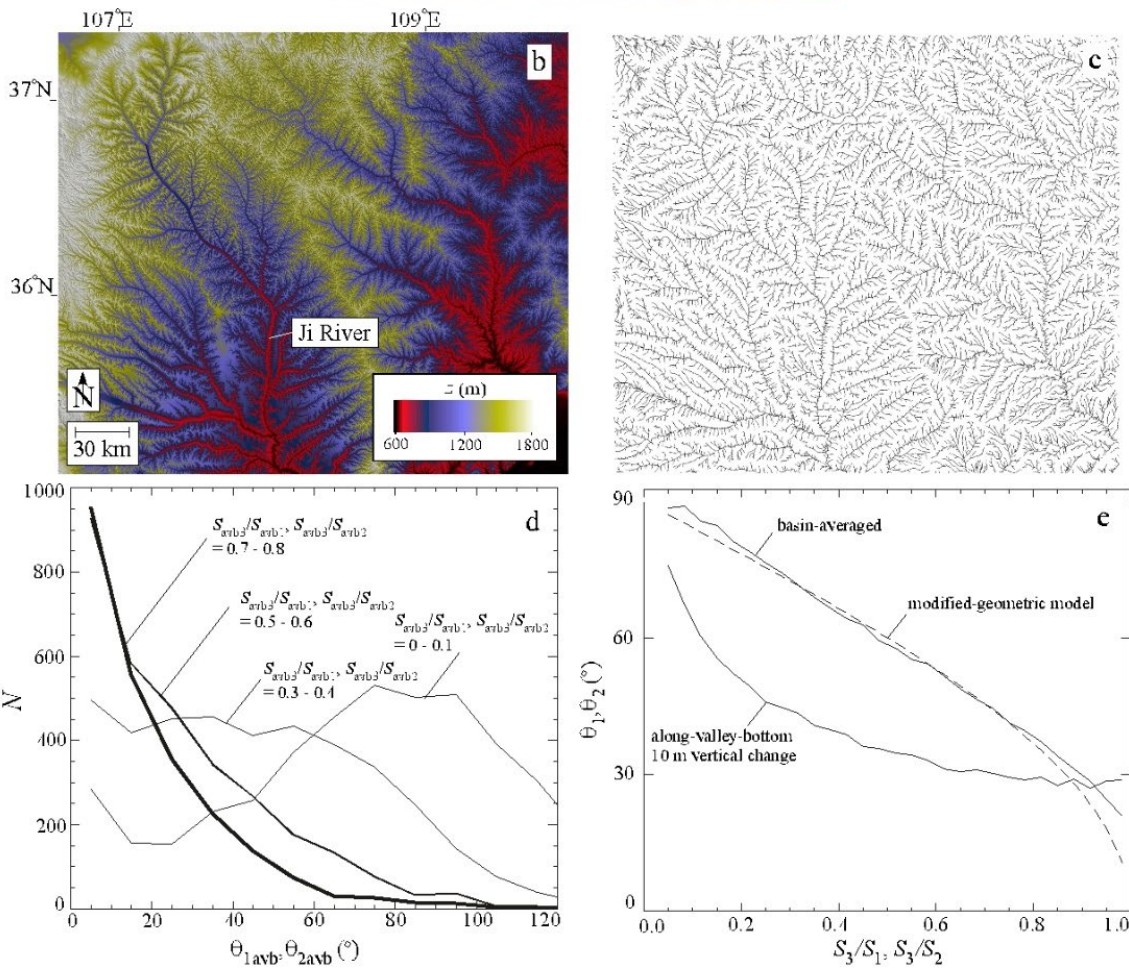

**Figure 9: Results of the junction extraction for a portion of the Loess Plateau region of China. (b) Color map of topography. (c) Valley bottom network extracted for the study area. (d) Plot of histograms of AVB junction angles for four ranges of slope ratios. (e) Plot of junction angles as a function of slope ratios for both BA and AVB properties.**



*3.1.5 Results for the conterminous U.S.A. (CONUS)*

Figure 10 illustrates the input data for the analysis of junction angles in CONUS. Figure 10a is a shaded relief image of the NED in LCC projection. The GitHub repository for this paper (Pelletier, 2024) includes an image of the entire drainage network map of the area extracted by the algorithm along with the positions, angles, and slope ratios of each of the 19,682,591 junctions. Figure 10b is a grayscale map of the surficial geologic map of Soller et al. (2009) simplified to three map units: 1) Plio-Quaternary alluvium and the Miocene-to-Pliocene Ogallala Formation, 2) bedrock and older alluvial

deposits, and 3) glacial/aeolian deposits. Our statistical analysis of tributary fluvial network junction angles presented here includes only the areas in white (late-Cenozoic alluvial piedmont deposits) and dark gray (bedrock/older deposits) to avoid the drainage contortions that may be associated with glacial/aeolian deposits.

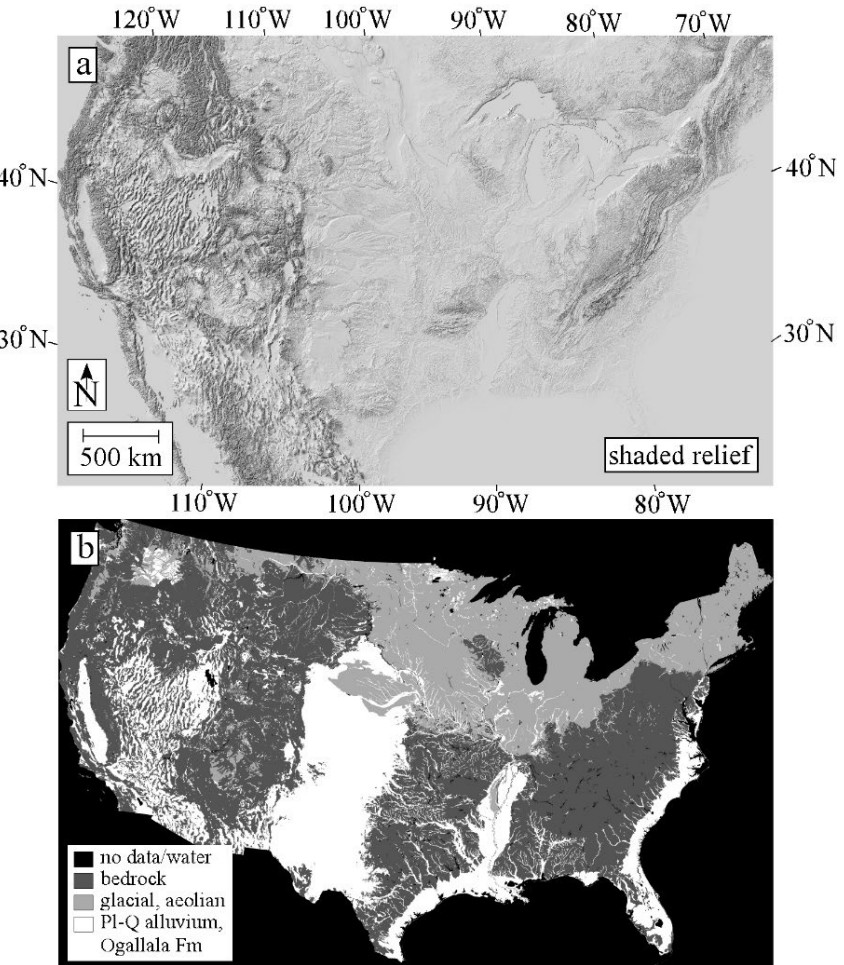

**Figure 10: (a) Shaded relief map of the conterminous U.S.A. in Lambert conformal conic projection. (b) Surficial geologic map of the conterminous U.S.A. (Soller et al., 2009) simplified to three units.**



Figure 11a is a color map of the geometric mean of all junction angles within each 2.5 m x 2.5 km square in CONUS. This figure illustrates that mean junction angles are commonly in the range of 35˚-45˚ in large parts of CONUS. Mean junction angles are generally lower, i.e., 15˚-25˚ in areas of late-Cenozoic alluvial piedmont deposits (i.e., the Ogallala Formation (Figs. 11b&11c) and in piedmonts of the western U.S.A. (Figs. 11d&11e)). The change in mean junction angle between relatively low values associated with late-Cenozoic alluvial piedmont deposits and higher values associated with bedrock/older deposits occurs abruptly at bedrock-alluvial contacts, not gradually as would be the case if climate were the primary control on junction angles (given that aridity changes gradually with elevation compared to the abrupt transition in presence/absence of late-Cenozoic alluvial piedmont deposits that occurs at mountain fronts). Junction angles can be relatively high, i.e., close to 90˚, along some of the major rivers of the Great Plains (e.g., the Platte, Republican, Arkansas, and Cimarron Rivers), similar to the pattern observed along the major valley-floor channels of southern Arizona in Figure 8 where two piedmonts of opposing orientation intersect.





**Figure 11: Color maps of junction angles averaged for every 2.5 x 2.5 km in the conterminous U.S.A. Junction angles tend to be lower in areas with late-Cenozoic deposits of (b)&(c) the Great Plains and (d)&(e) the Basin and Range province of the southwestern U.S.A.**



A Spearman correlation analysis for CONUS demonstrates that the mean junction angle computed at the 2.5 km scale has a
correlation coefficient with the presence/absence of late-Cenozoic alluvial piedmont deposits (Spearman correlation
coefficient of $\rho = 0.11$ and $p$ value of $<10^{-100}$) that is approximately fifty times higher than for the correlation coefficient with
the aridity index ($\rho = 0.002$ and $p$ value of 0.01). This analysis was repeated with drainage networks extracted using a
threshold area of 0.3 km$^2$. The Spearman correlation coefficient is essentially identical to the one obtained with the threshold
contributing area of 0.1 km$^2$, and the map of mean junction angle obtained with a threshold contributing area of 0.3 km$^2$ is
visually indistinguishable from Figures 11b&11d.

## 3.2 Synthetic landforms

### 3.2.1. Idealized branching tree test

The idealized branching tree used to test the junction angle extraction algorithm (Fig. 4) has two junctions of 45°, eight
junctions of 40°, two junctions of 35°, and fourteen junctions of 30°. The algorithm extracts the two 45° junctions with a
mean of 44.7° and a standard deviation of 0.3°, the eight 40° junctions with a mean of 39.3° and a standard deviation of 1.0°,
the two 35° junctions with a mean of 36.3° and a standard deviation of 0.8°, and the fourteen 30° junctions with a mean of
29.9° and a standard deviation of 0.9°.

### 3.2.3. Results of junction angle extraction for flow over tilted planar landforms with random microtopography

The networks defined by steepest-descent directions for flow over tilted planar landforms with random microtopography and
hydrologic correction illustrated in Figures 12a-12c transition from lower mean junction angles to higher mean junction
angles with increasing $S_l/S_r$. In Section 1, we estimated that $S_l/S_r \sim 1$ in incised late Cenozoic alluvial piedmont deposits,
hence the results in Figure 12a are most applicable to those portions of the landscape. We also estimated that $S_l/S_r$ is likely
$\gg 1$ in bedrock and older deposits that have been faulted or folded prior to or coeval with tributary drainage network
development. As such, the results in Figure 12b&12c are most applicable to those portions of the landscape. Figure 12d plots
the junction-angle histograms associated with the fluvial networks in Figures 12a-12c. Figure 12d demonstrates a systematic
increase in mean junction angle (indicated by the vertical dashed lines near the top of the graph) with increasing $S_l/S_r$. Figure
12e plots the mean junction angle as a function of $S_l/S_r$ for 30 different realizations of these synthetic landforms constructed
with a range of values for $S_l$, $S_r$, and $v_0$. Mean junction angles systematically increase with $S_l/S_r$ and are not sensitive to $v_0$.
Figure 12f illustrates the equivalent of Figures 8b&8d for flow over tilted planar landforms with random microtopography.
These results are similar to those for the landforms of southern Arizona and the Loess Plateau, i.e., junction angles computed
using BA properties closely follow the predictions of the MGM while those computed using AVB properties match the
predictions of the MGM for slope ratios near 0 and 1 but deviate from the predictions of the MGM for junction angles
associated with slope ratios that are mid-ranged.



**Figure 12: Results for flow over tilted planar landforms with Lorentzian microtopography. (a)-(c) Fluvial networks obtained with increasing values of $S_l/S_r$. (d) Plot of histograms of along-valley bottom junction angles for the fluvial networks illustrated in (a)-(c). (e) Plot of mean AVB junction angles as a function of $S_l/S_r$ for a range of values for $S_l$ and $S_r$. (f) Plots of junction angles as a function of slope ratios for BA and AVB properties.**



*3.2.4. Why do junction angles computed using BA properties follow the MGM while those computed using the MGM deviate systematically from the MGM?*

Previous sections have documented that the trends in mean junction angle versus slope ratio are different for BA and AVB properties (with data for BA properties consistent with the MGM and data for AVB properties deviating systematically from the MGM). We posit that AVB properties deviate from the predictions of the MGM due to local variations associated with

valley meandering/tortuosity. To test this hypothesis, we varied the scale over which AVB slopes and orientations are computed for southern Arizona using an elevation change over which valley-bottom slopes and orientations are computed of 5 m and 30 for comparison with the results obtained using the default value of 10 m. Figure 8c demonstrates that computing the AVB properties using a larger elevation change results in data that are more consistent with the predictions of the MGM compared to the results obtained using AVB properties computed over a smaller elevation change. BA properties represent

the end-member case of computing slopes and orientations using all points in a drainage basin, thus eliminating any randomness/variation due to local valley meandering/tortuosity. These results support the hypothesis that AVB properties deviate from the predictions of the MGM due to variations associated with valley meandering/tortuosity.

## 4 Discussion and Conclusions

### 4.1 Summary of key findings

Using a novel junction-angle extraction algorithm tested using an idealized branching network with known junction angles, we developed a database of $\sim 10^7$ junction angles for CONUS. Mean junction angles computed using basin-averaged properties are consistent with the MGM while mean local along-valley-bottom orientations and slopes deviate systematically from the MGM, a deviation that we propose is likely the result of variations in slopes and/or orientations associated with valley-bottom meandering/tortuosity. We mapped the spatial distribution of mean junction angles at 2.5-km scale and

documented systematically lower mean junction angles in locations of incision into late-Cenozoic alluvial piedmont deposits compared to incision into bedrock/older deposits. We posited that areas of late-Cenozoic alluvial deposition likely have a low initial ratio of mean microtopographic slope to the large-scale slope/tilt because alluvial deposition is associated with avulsion and topographic diffusion that, at analog sites such as the Holocene alluvial piedmonts of Ft. Irwin, are characterized by unusually low microtopography (i.e., ~1 m over spatial scales of ~100 m). We demonstrated that lower

ratios of mean microtopographic slope to the large-scale slope/tilt are associated with lower mean junction angles even before any fluvial incision takes place (Fig. 12).

### 4.2 Comparison of the junction angle dataset of this paper to NHDPlusV2

Several recent junction-angle studies (e.g., Seybold et al., 2017; 2018, Getraer and Maloof, 2021) are based on the

NHDPlusV2 dataset (McKay et al., 2012). The NHDPlusV2 dataset has two limitations/drawbacks that motivated our development of the alternative junction angle dataset for CONUS presented here. First, the NHDPlusV2 does not include



many small valleys visually apparent in shaded-relief DEMs of southern Arizona (Fig. 6c versus Fig. 6d). Differences in drainage density are apparent across political (e.g., county) boundaries in the NHDPlusV2 dataset, suggesting that the criteria used to define a valley in that dataset vary in a manner that depends on boundaries that are arbitrary from a
geomorphic perspective. Second, the digital topographic analysis used to estimate along-valley-bottom slopes in NHDPlusV2 includes a procedure that reverses negative slopes (McKay et al., 2012, p. 122). The algorithm of this paper obviates the need to enforce such slope reversals by computing the along-valley-bottom slope over a horizontal distance that automatically varies because it is based on a prescribed along-valley-bottom elevation change, hence areas of lower slopes necessarily have their slopes computed over a longer horizontal baseline that guarantees positive slopes.


Our approach of extracting tributary valley networks using a uniform threshold contributing area for identifying valley heads has drawbacks that we want to clearly acknowledge, including the potential for over-mapping valleys in some areas and under-mapping valleys in others. More advanced procedures for valley-network extraction use a threshold contour curvature rather than a threshold contributing area for identifying valley heads (e.g., Pelletier, 2012; Hooshyar et al, 2017). We chose
not to use such an approach in this study because drainage network extraction at 1 m pixel[-1] resolution for all of CONUS would be computationally difficult. We mitigated potential problems with using a uniform threshold contributing area for valley-head identification by demonstrating that the results are independent of the threshold contributing area value chosen within a reasonable range (0.1 to 0.3 km$^2$).

While we acknowledge the limitation of using a uniform threshold contributing area to identify valley heads, we also wish to note that the use of such a threshold does not result in uniform hillslope lengths because variations in the degree of topographic convergence translate into a range of hillslopes lengths even when a uniform threshold contributing area is used to identify valley heads. To see this, consider the difference in hillslopes lengths between a planar hillslope (oriented along a cardinal direction to simplify the example) with pixel size of 50 m versus a convergent hillslope, square-shaped in planform,
formed by the intersection (along the diagonal of the square) of two planes with aspects that differ by 90°. In the case of the planar hillslope valley heads will be identified (using a threshold contributing area of 0.1 km$^2$) at every pixel located 2000 m from the drainage divide because all flow pathways are parallel and 50 m x 2000 m equals the prescribed contributing area threshold of 0.1 km$^2$. The convergent hillslope of maximum contributing area of 0.1 km$^2$ has a maximum length along the diagonal of 447 m (i.e., the hypotenuse of an isosceles right triangle with legs of length equal to the square root of 0.1 km$^2$ or
316 m). As such, the use of a uniform threshold contributing area for identifying valley heads, while simplistic, nevertheless allows hillslope lengths to vary by approximately a factor of 4.

## 4.3 Comparison of results to prior studies

Seybold et al. (2017; 2018) demonstrated a correlation between mean junction angles and the aridity index in CONUS. Yi et
al. (2018) further related aridity to the drainage basin aspect ratio and Hack exponent. We suggest that the correlations that



Seybold (2017; 2018) and Yi et al. (2018) attributed to aridity may be more directly related to the presence/absence of late Cenozoic alluvial piedmont deposits, because both aridity and deposition depend on elevation, i.e., lower elevation areas are more likely to be both arid (e.g., Basist et al., 1994) and depositional.

The correlations between junction angles and their controlling parameters depend on the scale over which junction angles are averaged. Seybold et al. (2017; 2018) averaged all junction angles in each Hydrologic Unit Code 6 drainage basin (average contributing area of ~30,000 km$^2$) to arrive at a single value of mean junction angle against which aridity was compared. Averaging over such large scales results in junction angles from areas with bedrock/older deposits being lumped together with those of late Cenozoic piedmont alluvial deposits. Some averaging of junction angles is likely necessary when studying

the controls on tributary junction angles because averaging is helpful for identifying trends that may otherwise be obscured by the specific pattern of valley-bottom meandering/tortuosity near tributary junctions. We propose, however, that it is necessary to average junction angles over areas that are sufficiently small to resolve the presence/absence of late Cenozoic alluvial piedmont deposits (2.5 km is used here) given the fundamentally different nature of the initial topography in cases of drainage development into late Cenozoic alluvial piedmont deposits compared to drainage development into bedrock/older

deposits.

The climate-based model for junction angles is based on a two-step conceptual model in which 1) greater aridity results in less infiltration that, in turn, results in 2) increased erosion by surface water flows that cause fluvial valleys to align more closely with the large-scale slope/tilt (Seybold et al, 2017, p. 2278). Whether or how increased erosion rates cause fluvial

valleys to align themselves more closely with the large-scale slope/tilt is unknown, but more arid regions are not associated with less infiltration relative to precipitation. On a mean-annual basis, Budyko (1974) demonstrated that runoff coefficients are generally lower in more arid areas, indicating more infiltration and/or evapotranspiration relative to precipitation in such climates. On an event basis, which is likely the most relevant time scale for assessing erosional efficiency, runoff coefficients are sufficiently complex (i.e., dependent on the seasonality of precipitation, presence/absence of substantial

snowmelt runoff, etc.) that no clear relationship with aridity exists for all of CONUS (Stein et al., 2021). However, any tendency for hillslopes in areas of greater aridity to have higher runoff coefficients due to a prevalence for infiltration-excess overland flow is likely to be counteracted by the tendency of runoff coefficients in such climates to decrease as a result of the spatial variability of precipitation, large channel transmission losses (Simanton et al., 1996), and greater plant water-use efficiency (Troch et al., 2009).


Seybold et al. (2017) and Hooshyar et al. (2017) documented $\theta_1 + \theta_2$ values in the range between 45° and 72° in Seybold et al. (2017) and 49.5˚ and 75.0˚ in Hooshyar et al. (2017). Hooshyar et al. (2017) associated 49.5˚ with drainage networks incised primarily by debris flows. They inferred debris-flow dominance by assuming that the topographic signature of debris-flow-dominant erosion is a fluvial channel slope-area scaling exponent that is less negative (i.e., closer to zero) in





such drainage basins compared to those in drainage basins dominated by fluvial erosion. The results presented here could
      potentially be reconciled with those of Hooshyar et al. (2017) if tributary fluvial networks incised into late-Cenozoic
      piedmont alluvial deposits have slope-area scaling exponents with smaller absolute values (again, due to lower initial $S_l/S_r$)
      compared to those of incised bedrock/older deposits.

A persistent hypothesis in the junction-angle literature is that junction angles evolve toward a state of minimum-power
      expenditure (Strong and Mudd, 2022 and references therein). Howard (1990) (his Table 1) demonstrated that the predictions
      of such optimality principles are nearly indistinguishable from those of the MGM. Minimum-power relationships make
      similar predictions to those of the MGM because slope and contributing area/discharges tend to be inversely correlated in
      fluvial systems, hence the MGM-based relationship between junction angles and slopes also presents as a relationship

between junction angles and contributing areas/discharges (which relate to power expenditure). We view the debate about
      whether optimality principles or some more fundamental mechanism such as the MGM is the primary control on tributary
      fluvial network junction angles as analogous to the debate over how to interpret Horton's Laws for such networks. Horton's
      Laws have been interpreted to be a result of optimality (e.g., Rigon et al., 1993), but Kirchner (1993) proved that they are
      statistically inevitable given the branching architecture that results from Strahler ordering on a surface that is required to

drain an area through a point. Similarly, we propose that the MGM represents a fundamental/inevitable constraint that should
      be considered as a null hypothesis before less fundamental/inevitable principles are proposed as primary controls on junction
      angles.

### 4.4 Additional factors that may to contribute to the larger junction angles of fluvial networks incised into bedrock

The existence of an initial, pre-incision topography characterized by random microtopography superimposed on a large-scale
      slope/tilt is reasonable for late-Cenozoic alluvial piedmont deposits but less clearly applicable in cases of fluvial-network
      development into bedrock/older deposits. Such landforms may be shaped by faulting and folding that occurs over a broad
      range of spatial and temporal scales as well as by relief production via spatial variations in bedrock erodibility. By focusing
      our analysis on the flow that occurs on tilted planes with varying degrees of small-scale topographic roughness, we have left

out many potential mechanisms, particularly those in bedrock landforms, that may influence junction angles, including
      preferential erosion along vertically oriented joints (Pelletier et al., 2009), lateral tectonic advection (Hallet and Molnar,
      2001), etc. We emphasize the role of initial $S_l/S_r$ in this study because we believe that it is the most relevant factor for
      understanding the spatial variations in mean junction angles in CONUS, especially the difference between incised late-
      Cenozoic alluvial deposits and bedrock/older deposits. However, it is far from the only control on fluvial network junction

angles.

      It is also important to note that relatively low mean junction angles are found in some bedrock landscapes where drainage
      divides are unusually linear in planform and slopes are especially steep. Figure 13, for example, illustrates parallel and




subparallel drainage development in bedrock using the Cambrian sedimentary rocks (Wrucke and Corbett, 1990) of the Lost

Chance Range of California (Fig. 13a) and the granite and schist (Bryan, 1925) of the Mohawk Mountains of Arizona (Fig. 14b) as examples. The relatively low junction angles of such steep bedrock terrains are broadly consistent with the MGM because the relatively linear nature of the drainage divide in planform and the steep nature of the large-scale slope/tilt are likely associated with relatively low initial $S_l/S_r$ values in such cases than is typical in bedrock landscapes.

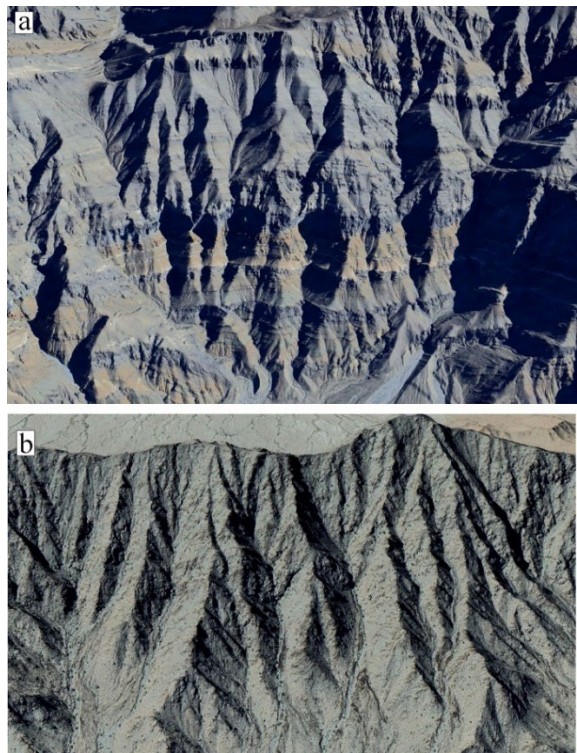


**Figure 13: Oblique aerial photographs of portions of subparallel drainage networks incised into bedrock in (a) sedimentary rocks of the Lost Chance Range, California, and the granite and schist of the Mohawk Mountains, Arizona.**

*Code/Data Availability* DEM data and codes used to extract junction angles are available at Pelletier (2024).

*Author contributions* JDP wrote the codes and the paper. RGH compiled the input data for CONUS. RGH, OH, and BF
participated in the analysis. JDP benefitted from discussions with LAM on this problem for the past 10+ years.

*Competing interests* The authors declare they have no competing interest.





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
