# Peer review of "Geometric constraints on tributary fluvial network junction angles"

_EGUsphere, 2024_

## Author Response (AR1)

**Response to referee #1:**

We thank referee #1 for their comments. We welcome the opportunity to clarify the issues they raised. We first summarize the main findings of our paper and then respond to several overarching criticisms offered by referee #1. We then address their specific line-by-line comments and detail what we have changed in the manuscript to address their concerns. We would appreciate the opportunity to make additional changes that the editor and/or reviewers deem necessary.

Our preprint argued that landscapes with slopes that initially tend to be aligned along the regional slope direction across a wide range of spatial scales tend to have lower mean junction angles. We established this based on a new high-resolution dataset for the conterminous U.S.A. that uses methods/codes that we have shared with the community and that we explicitly tested on synthetic branching landscapes to demonstrate their accuracy. We argued that the geomorphic history of alluvial piedmonts (including the specific tectonic and depositional processes that create alluvial piedmonts) lends itself to especially planar initial conditions (i.e., low $S_l/S_r$ values), consistent with the correlation we document between junction angles and the presence/absence of late Cenozoic alluvial deposits (which, prior to fluvial incision, likely tend to have lower $S_l/S_r$ values compared to bedrock/older deposits). As a result, we caution against using junction angles to infer paleoclimate (as was done for Mars by Seybold et al., 2018). Our argument is not based merely on a correlation between junction angles and the tendency of slopes to be aligned along the regional slope direction across a wide range of spatial scales, but instead on how two planes intersect to form a line and the associated equation for how junction angles are partially constrained by the ratio of slopes of the main valley bottom and each tributary valley bottom.

Referee #1 concluded that the essential elements of our study were previously published by Castelltort (2013) (which we take to be Castelltort and Yamato (2013) since no other paper dealing with drainage basins was published by Castelltort in 2013). However, Castelltort and Yamato (2013) did not study junction angles (they studied the length to width ratio of drainage basins). They did not extract any network. They did not reference Howard, whose geometric model is a critical part of our study. Our work was undoubtedly inspired by, and owes a large intellectual debt to, the work of Castelltort and Yamato (2013). We tried to make that intellectual debt clear in the caption to Figure 2 of our preprint where we wrote "The specific examples in this figure are ours but the concept closely follows Castelltort and Yamato (2013)."

Please also note that the data for real landscapes that we present tell nearly the opposite story to that of the data for real landscapes presented by Castelltort and Yamato (2013) (their Figure 2). Their Figure 2 plots the convergence angle (a measure of the aspect ratio of a drainage basin) versus regional slope, illustrating that higher regional slopes result in more elongate drainage basins. What we demonstrated in our preprint is very different, i.e., that tributary fluvial networks incised into alluvial piedmonts (which tend to have lower regional slopes than tributary fluvial networks incised into bedrock/older deposits) have lower mean junction angles than those incised into bedrock/older deposits. We proposed that this trend exists because, while the regional slope is

lower for alluvial piedmonts compared to bedrock landscapes (which, all else being equal and if $S_l/S_r$ values are a control, would result in larger (not smaller) mean junction angles on alluvial piedmonts), the initial microtopographic slope is far lower still (the distributary Holocene alluvial piedmonts of Fort Irwin that we studied provide a modern analog for the initial conditions of late-Cenozoic alluvial piedmont deposits that are now incised and have tributary fluvial networks developed in them) with the result that alluvial piedmont deposits have lower mean junction angles *despite their lower regional slopes* compared to tributary fluvial networks incised into bedrock/older deposits.

Referee #1 states that slopes must be a result of geomorphic processes that occur during valley incision, including erosion, deposition, and tectonic uplift. We couldn't agree more. That fact does not preclude the geometric model and/or initial flow pathways in the landscape from playing a substantial role in controlling fluvial network junction angles. Clearly, tectonic uplift, substrate erodibility, and erosional and depositional processes do control slopes, but it is *the ratio of slopes*, not their absolute values, that is the independent variable (i.e., the variable controlled by other factors) in the geometric model for tributary junction angles (the dependent variable in the geometric model). As such, the geometric model for junction angles is not inconsistent with any proposed control on slopes. The conceptual model we are proposing includes geomorphic processes explicitly, albeit potentially different ones than reviewer #1 believes control junction angles. One example: lines 68-75, in which we described the specific tectonic and depositional processes that create alluvial piedmonts and their tendency for slopes to align, across a wide range of scales, with the regional slope. Referee #1 rejects our argument that alluvial piedmont deposition creates low-relief landforms, a point that we have worked hard to address (see response to L40 comment below).

How might a more parallel initial drainage architecture tend to be preserved as valleys incise? One possibility is that more parallel and elongate initial valleys have relatively similar contributing areas (relative to the power-law distribution of contributing areas typically observed in drainage networks with a more dendritic architecture, e.g., Veitzer et al., 2003), leading to relatively similar incision rates among neighboring drainage basins, thus tending to maintain the parallel nature of the landscape by minimizing the tendency for one or a few drainage networks to incise faster than others. We have included new numerical modeling work (see new text in the response to referee #2) designed to show how both the initial condition and geomorphic processes of incision combine to control junction angles.

We respectfully disagree with the referee that the correlations documented in our paper are "spurious." In documenting a correlation between mean junction angles and the presence/absence of late Cenozoic alluvial piedmont deposits, we have taken the analysis as far as it will go because the initial $S_l/S_r$ value of a now-incised landscape cannot possibly be measured. In such cases, the best one can do is establish a correlation with a quantity or characteristic (i.e., the presence/absence of late Cenozoic alluvial piedmont deposits) that is related to the property that cannot be measured (i.e., the initial $S_l/S_r$ value of a now-incised landscape) and then make a process-based association

between that quantity/characteristic and the property that cannot be measured. That is what we have done.

We regret that our writing came across as non-inclusive. We intended to propose that the ratio of the initial microtopographic slope to the regional slope, $S_i/S_r$, is a constraint (not the only constraint) on junction angles. The preprint contained a paragraph emphasizing that $S_i/S_r$ is one of many controlling factors (lines 630-640). Since the preprint did not convey this message as intended, we have rewritten the introduction and discussion to further clarify that many processes and factors likely control junction angles.

Line-by-line comments and responses:

"While the paper makes some interesting points on how topographic features and geology may be expressed in a channel network's branching geometry, several points need to be clarified and some miss-perceptions of previous studies need to be resolved. First, the fact that vertical features of topography like slope correlate with horizontal geometric properties of landscapes such as branching angles should surprising. Ultimately, Earth's surface evolves as a result of erosion and uplift and valley networks are simply characteristic features embedded in this three-dimensional topography. More important than correlations between different topographic features is the understanding of how the underlying processes, namely erosion, deposition, and tectonic uplift are expressed in the network's branching structure. Quoting from Howard's 1990 paper on the grometric model: , "[this geometric model] was only sketchily related to the assumed channel processes ...". This point actually prompted Howard in 1990, to explore additional hydrological arguments to constrain the angle-slope relation foundon more process-based descriptions of how the flow in a channel may modify its bed, even if the particular expression may result in a slightly lower r-square, than the rpedictions of the geometric model. Also note, that much of the theoretical basis of Howard 1990 was mostly done 20 years earlier in [Howard 1971b, 1971c] as referenced in the 1990 paper. Similarly, Hooshyar 2017 and Seybold 2017 try to relate branching angles to channel-forming processes, namely overland flow v.s. debris flow in the first and seepage flow v.s. overland flow in the latter, while the optimality models of Strong & Mudd e.g. use hydrological conservation arguments at the junction, similar to the arguments in Howard 1990. I would like to note here, that while the authors promote the impression that the observed slope-angle relation contradicts the above-mentioned interpretations, however, these conceptual frameworks are more complementary as they simply provide different aspects for explaining how landscapes evolve. Consequently, the authors may consider a more "inclusive tone" in their argumentation, particularly because the author's own explanations remain rather vague on process understanding giving too much focus on the trivial angle-slope relations of the geometric model."

Response: The referee is correct that Howard (1990) did not find the geometric model to be compelling as a process-based understanding of junction angles. That does not preclude others from arguing that the geometric model and/or initial flow pathways exert some control on the

junction angles of tributary fluvial valley networks. We regret that we did not use a more inclusive tone.

"An interesting point in the manuscript which I was initially excited to read about was the alleged finding that incisions into alluvial piedmonts exhibit different structural controls on branching angles than channel networks incised into bedrock. Here, the role of geology is an important point for channel erosion and has not been addressed adequately in previous work, including the recent papers cited by the authors. However, the more I was disappointed when reading through the text to find out that the narrowing of branching angles in alluvial piedmonts is fully explained by the fact that piedmonts are (usually) preferentially sloped while bedrock valley networks do not display a preferential slope. Thus, what is thought to be a structural effect is simply the higher prevalence of regional gradients that narrow the branching angle statistics. This effect of regional slope and microtopography has been already demonstrated by Casteltort in 2013 through numerical simulations and also observed in multiple previous studies over the last decade."

Response: Thank you for this comment. As noted above, Castelltort in 2013 did not study junction angles nor any other aspect of the network (they studied the aspect ratio of drainage basins).

"L.5: Ignoring some degenerated special cases, three planes, one for each upstream tributary and one for the downstream channel can intersect at any angle and slope and thus does not provide a geometric constraint for branched valley networks. What actually constraints the argument of Horton /Howard is the presence of fixed a general slope (for all three basins) aligned with the downstream channel, while the side valley is additionally inclined with respect to this dominant direction."

Response: Thank you for this comment. The geometric model for tributary fluvial network junction angles does not involve three planes (intersecting or otherwise). The geometric model of Howard (1990) posits that two planes (approximating the topography of the tributary valley bottoms and the topography upslope from and surrounding those tributary valley bottoms) *intersect to form a line* that constrains the orientation of the main/downstream valley bottom.

"L 9-10: Landscapes evolve through erosion/deposition and uplift creating sloped valley networks. Thus I firmly object to the implications that the geometric "model" actually contradicts the process-based explanation by Hooshyar et al. or Seybold et al. as well as the optimality principles proposed by Rinaldo, Strong & Mudd, and many others. Earth's surface is shaped by processes and not geometric features."

Response: Thank you for the opportunity to clarify our thoughts on this aspect of our work. We concluded that the initial condition prior to incision is one "constraint" (the title of the paper, line 26, and elsewhere) on junction angles. Also note that, as discussed above, the ratio of slopes can be viewed as the independent variable in the geometric model, hence that ratio can be controlled by any process with no contradiction with the geometric model for tributary junction angles. No text in our preprint precluded other processes from being important controls on junction angles.

We did include a sentence proposing that the geometric model be used as a null hypothesis to be disproven before other hypotheses are accepted. We have removed that sentence from our paper to avoid the unintended impression that we believe that any of our work contradicts other proposed models for junction angles.

L. 11-12: "Junction angles are consistent with the geometric model..." It would be good to have a 1:1 plot between prediction vs measurement and a quantification of the spread and uncertainty of the model prediction. Also, it would be favorable to see how the "classical geometric model" compares to the "modified geometric model" in such a comparison.

Response: Thank you for this comment. We have modified Figure 8 of our preprint to plot the observed versus predicted junction angles (see revised Figure 8c&8d below). The classical geometric model of Horton (1932) is counterfactual because it assumes that the larger of the two tributaries is precisely the same orientation as the main (downstream) valley. It was superseded 52 years ago by a more realistic model, and it will not be considered in this study.

[Figure]

**Revised Figure 8. Plots illustrating the relationships of junction angles to the ratios of slopes downslope and upslope of the junctions for the portion of southern Arizona illustrated in Figure 5. (a) Plot of junction angles measured using BA properties as a function of the ratio of slopes downslope and upslope. (b) Plot of junction angles measured using AVB properties as a function of the ratio of slopes downslope and upslope. (c)&(d) Plots of observed junction angles versus those predicted by the MGM using the data of (a)&(b), respectively. (e) Plots of junction angles measured both AVB (using three different values of the elevation change over which slopes and orientations are computed) and BA properties.**

To put the scatter in the revised Figures 8c&8d into context and to mitigate concerns that we are using a new junction-angle dataset, Figure R1a below plots the relationship between the measured sum of junction angles $\theta = \theta_1 + \theta_2$ from the classic NHDPlusV2 dataset and the predictions of the geometric model using the supplementary data of Getraer and Maloof (2021) (https://hydroshare.org/resource/0b93e7c659fe4fc59bf6a202c269313c/) and the relationship between the measured sum of the junction angles and the logarithm of the aridity index using the supplementary data of Seybold et al. (2017) (grl55580-sup-0001-supinfo.csv). A key point of Figure R1 is that there is substantial scatter when plotting individual (unaveraged) junction angles regardless of the data source used and the model or controlling variable being proposed (geometric vs. aridity-based).

[Figure]

**Figure R1. Relationships between (a) the observed sum of the two junction angles, $\theta = \theta_1 + \theta_2$, and the predicted value according to the geometric model, using the supplementary data of Getrear and Maloof (2021) and (b) the observed value of $\theta$ and the logarithm of the aridity index using the supplementary data of Seybold et al. (2017).**

"L.17-25: How does the type of geology (alluvial vs bedrock) relate to the sloping of the topography? The tilting effect on branching angles and basin shapes observed in the modeling procedures of Casteltort 2013) which the authors draw to explain their findings makes absolutely no assumption on the ground surface material. Neither piedmonts nor bedrock is part of Casteltort's model. Consequently, I see nothing new in the interpretation besides the finding that alluvial piedmonts at mountain fronts often display a general sloping tilt and thus have narrower branching angles on average. However, this has been known for over 10 years (Casteltort 2013) and also observed in multiple network studies."

Response: Thank you for this comment. Castelltort and Yamato (2013) did not study junction angles.

"L. 27: "demonstrates independent[ce] of climate and optimality principles": How can the authors ensure that the generation of slopes in landscape evolution is not the effect of climate-driven erosion/deposition processes (rainfall leads to discharge and channel incision as a consequence) which follow optimization rules of landscape evolution?"

Response: Thank you for this comment. We did not intend to suggest that the generation of slopes in landscape evolution is not the effect of climate-driven erosion/deposition processes. Line 27 merely stated that the geometric principle that we argue is a constraint on fluvial network junction angles is independent of climate. *It is the geometric principle that it is independent of climate*, not landscapes themselves. In any case, we removed this sentence.

"L 35: parallel and sub-parallel networks in piedmonts v.s. dentritic and rectangular basins incised in bedrock: Is this a geological effect (alluvium vs bedrock) or simply an effect that the piedmont has a dominant regional slope?"

Response: This is an effect of the low $S_l/S_r$ of the initial piedmont landscape following a phase of aggradation. See the response to the next comment for additional information.

"L. 40: Piedmont deposition: I don't buy the claim that the deposition cycles created initially unincised low-relief landforms because the timescales for deposition are rather similar to the timescales of valley incision. Thus one can hardly separate the two processes as they occur simultaneously. Here tha authors need to provide some additional arguments which support their claim."

Response: Thank you for this comment. We welcome the opportunity to clarify this important point.

There is abundant evidence from the Basin and Range province of the U.S.A. (which has more alluvial piedmonts than any other region in the U.S.A.) that deposition and erosion take place in separate phases on alluvial piedmonts, such that alluvial piedmont deposits are emplaced during a

depositional phase that culminates in a low-relief landscape, followed by an erosional phase during which tributary fluvial drainage networks develop.

First, it is important to note that alluvial piedmont deposits have a range of ages but deposits that pre-date the mid-Holocene period are the most common on many piedmonts. To see this, one can examine any geologic map of the Basin and Range province that recognizes multiple surficial geologic map alluvial units (as opposed to lumping them all into a single Tertiary and/or Quaternary alluvial map unit). A good example of such a map is Workman et al. (2002) (https://pubs.usgs.gov/mf/2002/mf-2381/mf-2381-a/mf-2381-a.pdf). All of the alluvial map units (labeled with a small "a") other than Qay are mid-Holocene or older in age. Note that such map units are the vast majority of the total area of the alluvial piedmonts in this map. These ages represent the time since substantial alluvial deposition last took place at each location.

Christensen and Purcell (1985) documented that alluvial piedmont deposits that are Pleistocene in age or older are undergoing a phase of erosion and have had sufficient time for the distributary drainage networks characteristic of active/late Holocene portions of the piedmont to be converted (via fan-head entrenchment and abandonment of the surface as a zone of widespread deposition to one of fluvial incision) into tributary fluvial networks. The finding that Holocene deposits tend to have distributary drainage networks while Pleistocene and older deposits have tributary drainage networks has been corroborated by many studies, the best known of which to us are the many papers of the Arizona Geological Survey that have mapped alluvial piedmonts in southern Arizona and have used the distributary versus tributary nature of the drainage architecture as an age indictor (along with many other criteria including absolute dating methods). Geomorphologists working in the Basin and Range province have long used soil development as an age indicator because soil development requires a stable surface (neither widespread or substantial fluvial erosion or deposition on the surface). For example, Dempsey et al. (1993) state: "If an alluvial surface has been stable for thousands of years or more, soil fabric obscures the original sedimentary fabric… Significant soil development requires that a surface be quite stable and not subject to frequent flooding." Soil development would not be an accurate age indicator if, as the referee concludes, alluvial piedmont deposits undergo fluvial erosion and deposition simultaneously.

We ask the editors to advise us on how to proceed in addressing this comment. We believe that the work of Bull (1991) adequately details the concepts we have briefly summarized in the last two paragraphs, hence our referencing of Bull (1991) in the preprint on the cyclic nature of deposition and fluvial incision on alluvial piedmonts is adequate. We are, however, open to expanding our discussion of this point if it is deemed necessary.

"L.56-58: If this is true, then the narrow branching angles in piedmonts are not the result of the underlying geology but a spurious effect of the gently sloping terrain. This effect has been very clearly demonstrated by Casteltort using landscape evolution models which are completely independent of the underlying geology."

Response: Thank you for this comment. We respectfully propose that it is inaccurate to characterize this portion of our analysis as a spurious correlation and we note again that Castelltort did not study junction angles. We are using the presence/absence of late Cenozoic alluvial deposits as a proxy for the tendency of the slopes of alluvial piedmont deposits to be aligned, across a wide range of spatial scales, with the regional slope direction. As stated earlier, we have carried the data as far as they will go because the initial $S_l/S_r$ value of a now-incised landscape cannot possibly be measured. It is common in geomorphology that some important factor of the initial landscape cannot be measured because e.g., the rocks or sediments have been removed or the landscape has otherwise evolved from some past condition that is hypothesized to control the present morphology. In such cases, the best one can do is to establish a correlation with a quantity or characteristic that is related to the property that cannot be measured and then make a process-based association between that quantity/characteristic and the property that cannot be measured. That is what we have done.

L63: "The authors fail to demonstrate that tortuosity is indeed the driver for wider junction angles compared to the arguments of Howard 1990. "may promote narrower junction angles ..." is simply not enough for a central result in a scientific publication.

Response: Thank you for this comment.

It is difficult to respond to this comment because we don't know where/how the referee believes that our analysis falls short. We showed that along-valley tributary junction angles approach the predictions of the geometric model as the horizontal distance over which the slopes and orientations of the valley bottoms were computed is increased. Smaller horizontal distances will be more affected by tortuosity/local variations in valley-bottom orientation and that larger horizontal distances will be less affected by tortuosity/local variations in valley-bottom orientation. Referee 2 agrees with our point that Lazarus and Constantine (2009) proved that increasing values of $S_l/S_r$ increase valley tortuosity. Figure 12a-c of the preprint demonstrates visually that greater tortuosity is associated with larger junction angles.

"L 104: the importance of piedmont deposits: As the authors have already explained in a previous paragraph, it is not the piedmont deposit (geology) that narrows the branching angle but the gentle regional slope that often comes with the depositional landscape at mountain fronts. Consequently, any sloped geology e.g. volcanic bedrock in Hawaii would also lead to the same effect."

Response: Thank you for this comment. The piedmont nature of the landscape does matter in that piedmont deposition leads to a relatively planar, tilted initial surface, partly as a result of processes that are specific to piedmonts, i.e., the topographic diffusion associated with aggradation (Pizzuto, 1987) and the tendency of avulsions to fill in low spots on the piedmont. These processes are very different from those present in Hawaiian bedrock volcanoes (the $S_l/S_r$ values of which we did not study and don't feel qualified to comment on).

"L. 133: How do the authors define the basin averaged slope in the case of n-th order junctions with n>1? Do they follow the longest upstream tributary up to its source, or do they average the slope of the whole upstream network tree? This point needs to be clarified."

Response: Thank you for this comment. Clarified: "When computing BA properties, the algorithm computes the average orientation and slope of every pixel whose outlet is that junction, using every pixel upslope along directions 1, 2, and 3 (the latter being the total area comprising drainage basins 1 and 2)."

"L 190: Same as above. Do the authors use the pixel-by-pixel slope of the DEM and then average over all pixels of the upstream basin? What is the upslope along directions? These points need to be clarified."

Response: Thank you for this comment. Clarified. We use the pixel-by-pixel slope of the DEM and then average over all pixels of the drainage basin.

"L. 229-231: "... suggests that junction angles may be systematically lower, on average, in fluvial networks incised into late-Cenozoic alluvial piedmont deposits than those incised into adjacent areas of bedrock/older deposits." The authors need to show that the structural control on junction angle is not a spurious slope bias in gently sloped depositional landscapes such as piedmonts."

Response: Thank you for this comment. The referee's insistence that we identify a "structural" control on junction angles may stem, in part, from their rejection (see their L40 comment above) of our argument that deposition and erosion take place in separate phases on alluvial piedmonts. We hope that the additional justification we have offered (see above response to their L40 comment) is compelling.

"L. 316: comparison with NHD: Comparing USGS's medium-resolution blue lines with a fixed flow accumulation threshold extracted channel network is not a fair comparison, particularly as one can obtain any drainage density by adjusting the drainage area threshold. NHDPlus (and its higher resolution companion NHDPlusHR have been extensively ground-checked). For example, I was unable to identify many of the fine channel networks from Fig. 6b/d around 32.3N/110.9W on aerial images on Google Maps."

Response: Thank you for this comment.

We have eliminated all criticism of the NHDPlusV2 dataset. The revised version of Figure 6 is

[Figure]

**Revised Figure 6. Comparison of the tributary fluvial valley bottom networks for the larger Tucson region. (a) Shaded-relief image of the 30 m pixel$^{-1}$ National Elevation Dataset (NED). Fluvial valley-bottom networks obtained in this study using (b) 30 m pixel$^{-1}$ NED data and (c) using 50 m pixel$^{-1}$ NED data.**

The referee refers throughout their review to the "channel" network. The referee's statement that they cannot find channel networks in our dataset might be explained by the fact that we are studying a different landform than the one they are looking for. Channels are defined as zones of localized water flow *with banks*. As stated throughout our preprint, we are studying valley bottoms, i.e., zones of localized water flow with or without banks. Our preprint used the word "channel" in only one context, i.e., in reference to the large (contributing areas of $> 10^4$ km$^2$) valley-bottom channels where piedmonts of opposing orientation intersect. Otherwise, we were careful to state that we are studying valley-bottom networks, i.e., the entire fluvial network up to the point where hillslopes become convergent and transition into valley bottoms (with or without banks). If one confines oneself to the channel network only, many of the smaller valley bottoms in the fluvial system will be missed.

The referee makes a good point by noting that another dataset (NHDPlusHR, first available in a non-beta-version on July 7, 2022) became available after our work was finished. Our choice not to use the NHDPlusHR dataset was partly an issue of timing: the dataset did not exist as an official

release when we completed our work in Spring, 2020 semester (the study was performed as part of a class project during the Spring 2020 semester at the University of Arizona, although it took another four years to finalize the wording of the preprint). In addition, the NHDPlusHR dataset is relatively impractical to use for a nationwide analysis because it is distributed as many files with a total size of 952Gb. In contrast, our nationwide network was distributed as a single image with size of 216Mb. Finally, NHDPlusHR vastly undermaps the valley network (much less so than NHDPlus, but still by a lot) relative to our dataset in some areas and on alluvial piedmonts especially. Figure R2 below shows a shaded-relief image of a typical study area comprised of late-Cenozoic alluvial piedmont deposits and a portion of the adjacent mountain range (in this case the northwest side of the Tortolita Mountains near Tucson) (Fig. R2a) along with the network maps extracted using our methods (Fig. R2b) and those of NHDPlusHR (Fig. R2c). The corner coordinates are in UTM zone 12. Both our dataset and NHDPlusHR vastly undermap the many thousands of fluvial valley bottom segments that are visually apparent in the shaded-relief map (the undermapping partly results from the fact that the shaded-relief map was made from 1 m/pixel lidar-based DEMs while lower-resolution DEMs were used input for the valley-network extraction for both our dataset and NHDPlusHR). Although both our dataset and NHDPlusHR undermap the valley-bottom network in this example area, our dataset does a significantly better job at capturing the smaller valley bottoms.

We acknowledge that Figure R2 is based on only one example area. Figure R2 does not prove that our dataset is superior to NHDPlusHR generally in its resolution of small valley bottoms, let alone other metrics. That said, the example area depicted in Figure R2 is representative of the general tendency of our dataset to capture more small fluvial valley bottoms on alluvial piedmonts.

478000E, 3604000N

[Figure]

490000E, 3596000N

**Figure R2. Maps of (a) shaded relief, (b) the fluvial network extracted by our methods, and (c) the fluvial network in the NHDPlausHR dataset for a portion of a typical piedmont comprised of late-Cenozoic alluvial deposits and a portion of the adjacent mountain range. The corner coordinates are in UTM zone 12.**

"L. 579: I agree with the AI~elevation argument which should be easy to test. (correlation elevation~AI and correlation angle~elevation"

Response: Thank you for this comment.

We added the Spearman correlation coefficient and the statistical significance of the relationships between elevation and aridity (significant) and between elevation and the presence/absence of late Cenozoic alluvial piedmont deposits (very significant).

We did not find a statistically significant correlation between junction angles and elevation for southern Arizona. We attribute this to two possibilities: 1) the fact that the low-elevation Santa Cruz and similar valley-bottom channels of the region have junction angles close to 90° may preclude a simple correlation between mean junction angles and elevation, and 2) the fact that, while the elevation of alluvial piedmonts is always low relative to their adjacent mountain ranges, the elevation of piedmonts varies substantially from one basin to another (alluvial piedmonts exist at elevations ranging from 600 to 1500 m a.s.l. even in the relatively small southern Arizona study site). Any correlation between elevation and junction angles, therefore, would have to exist despite the potentially offsetting effects of higher mean junction angles in low-elevation portions of some bedrock mountain ranges (e.g., 700 m a.s.l.) and the lower mean junction angles in some relatively high elevation (e.g., 1500 m a.s.l.) alluvial piedmonts.

New text: "Both aridity and deposition depend on elevation (the Spearman correlation coefficient between elevation and aridity in the southern Arizona study area is $\rho = 0.034$ and $p = \sim 10^{-5}$ and between elevation and the presence/absence of late Cenozoic alluvial piedmont deposits is $\rho = -0.40$ and $p < 10^{-100}$) with lower elevation areas being more likely to be both arid (e.g., Basist et al., 1994) and depositional."

"L 615: I agree with the author's similarity arguments between minimum power and MGM, but while MGM only relates different aspects of Earth's tomography with each other, the optimal branching models seek to explain WHY the slopes are as they are from a physical mechanistic perspective of landscape evolution in general and channel erosion in particular.  Here, the topographic slope itself is only a feedback variable for flow accumulation in the erosional term of the landscape evolution equation."

Response: Thank you for this comment.

As noted above, the ratio of slopes can be viewed as the independent variable in the geometric model, hence the geometric model is not inconsistent with any proposed control on slopes. This is also true of optimality principles: slopes can be viewed as the input to those models and junction angles are the output.

While it is not central to this rebuttal, we respectfully state that we do not believe that optimality principles are inherently superior to the geometric model because they are more process-based. The intersection of two planes is also a process, one that should not be dismissed just because it

does not explicitly relate to sediment transport, bank retreat, or some other fluvial geomorphic process. We are not aware of any paper based on optimality principles that explains why geomorphic systems should evolve to minimize stream power, momentum balance, etc. We accept that fluvial systems may evolve to minimize *gradients* of stream power as part of a maximum-entropy thermodynamics principle (e.g., Schneider and Sagan, 2005), but our understanding is that published studies of optimality principles applied to the fluvial system do not rely on such principles but instead assume that fluvial systems evolve to states of minimum power, momentum balance, etc. rather than demonstrating how or why they evolve to such states. The unanswered question is whether power minimization is a causal explanation for junction angles or a coincidental consequence of some other controlling process or principle.

The key point for this response is that we did not intend for our text to imply that optimality principles are ruled out by anything we have done. We still believe that the geometric model is an appropriate null hypothesis to consider, but we have removed the sentence that advocated consideration of the geometric model as a null hypothesis so as not to state or imply that the geometric model should be in any way favored over alternative explanations for junction angles.

**Response to referee #2:**

"I've read the paper with much interest and found it extremely well-written and fascinating. Only sometimes did I find myself a bit lost in the lengthy technical discussions, but this might be entirely my fault, as I am not an expert on the topic discussed here. Also, I appreciate that the literature and scientific debate on junction angles in tributary fluvial networks are extensive, so previous arguments need to be discussed in detail.

Being not an expert, I'll refrain from delving into the debate regarding what drives the distribution of junction angles and how this study copes with or does not cope with earlier literature. The pros and cons of the approach are discussed in sufficient detail, and results are put into perspective with previous literature in a way that allows even a non-expert reader to navigate through the discussion, although it is very technical.

Still, I'd like to raise two possibly minor comments, one being purely terminological and the other perhaps more fundamental.

First, I'd suggest the authors avoid using the term "meandering" to describe sinuous valley patterns. Meandering refers to landforms that grow orderly rather than randomly. The fact that sinuous patterns can arise even from unordered growth, and that at low sinuosity random versus ordered shapes are not distinguishable (Limaye et al. 2021)."

Response: Thank you for this comment. We agree that the word meandering is potentially misleading in this context and have removed it.

"Second, and more importantly, the major limitation I see in the idea behind the paper is that valley sinuosity is assumed to remain fixed once their shape and sinuosity have been determined by the initial microtopography (in the way Lazarus & Constantine's idea illustrates).

Kwang et al. (2021) have demonstrated how lateral erosion/incision of rivers into bedrock is fundamental for the development of dendritic drainage networks, to the point that without lateral erosion, landscape evolution models cannot turn non-optimal, non-dendritic networks into optimal dendritic ones such as those observed in nature.

For the point being made in this paper (that valley tortuosity matters when it comes to junction angles) I think lateral erosion should be considered/discussed because it has the potential to:

Alter river valley tortuosity over time, thereby impacting junction angles.

Cause the system to lose memory of initial conditions (inducing persistent reorganization in dynamic steady-state network shape), thus critically diminishing (if not completely erasing) the effect of initial microtopography on which the authors anchor all their analyses and results.

In short, assuming that river networks evolve from initial conditions to a frozen state with no further modifications poses a strong limitations to the new perspective the authroes bring abotu regarding the drivers of junction angles in fluvial tributary networks. Later network modification by lateral river erosion into valleys, which can induce major drainage reorganization and long transience in landscape shape, should be discussed more thoroughly because it's potentially tied to how tributary channels intersect each others.

Response: Thank you for this comment.

We have included new landscape evolution modeling aimed at demonstrating how geomorphic processes may be affected by initial conditions such that both initial conditions and geomorphic processes may exert control on fluvial tributary network junction angles (we use the word "may" because any landscape evolution model is a gross approximation to the complex dynamics of real landscapes). Our LEM includes lateral migration of both ridges and valleys in that the diffusive term representing hillslopes processes will drive lateral migration in any asymmetric valley or ridge. That said, we accept Kwang et al.'s point that lateral migration driven by fluvial processes may be dominant sources of lateral migration in bedrock landscapes, and may be underestimated by our model because fluvial in our LEM is assumed to occur vertically.

New text:

"*2.3.3 Landscape evolution model results with planar tilted landscapes as initial conditions*

The tilted planar landscapes with random microtopography described in Section 2.3.2 were input into a standard coupled hillslope-fluvial detachment-limited landscape evolution model described by Pelletier (2013) to study junction angles on landscapes with and without geomorphic evolution (results presented in Sections 3.2.4 and 3.2.3, respectively). The hillslope diffusivity $D$ was prescribed to be 10 $m^2$ $kyr^{-1}$ and the bedrock erodibility $K$ was chosen

to be 0.001 kyr$^{-1}$ because these values result in landscapes with a reasonable drainage density (~0.01 m$^{-1}$). $K$ values that are too low relative to $D$ can fail to develop fluvial channels and those that are too high can result in landscapes with fluvial valleys that extend to every pixel in the model domain. The models were subjected to uniform uplift of 0.1 m kyr$^{-1}$ relative to the base level at the lowest side of the square domain for 5-10 Myr, i.e., sufficient time for the landscape to reach an approximate topographic steady-state condition."

and

"3.2.4. *Results of junction-angle extraction for uniformly uplifted landscapes at steady state*

Figure 13 illustrates the steady-state topography output by a landscape evolution model with initial topography corresponding to landscapes whose fluvial networks are illustrated in Figures 12a-12c.

Figure 13d illustrates that, for the lowest value of $S_l/S_r$ illustrated in Figure 13a, the junction angle distribution is bimodal. Deep incision of the major valleys that are aligned parallel to the large-scale/regional slope triggers the development of steep low-order tributary valleys that join with the main valleys at junction angles close to 90°. Larger values of $S_l/S_r$ have sufficient small-scale roughness that major slope-parallel valleys do not form and the junction-angle distributions are unimodal. As in the results obtained for flow over tilted planar landscapes with random microtopography in Figure 12, mean junction angles increase with increasing $S_l/S_r$ (dashed vertical lines in Fig. 13d).

[Figure]

**Figure 13: Results of landscape evolution models using the landscapes whose fluvial networks are illustrated in Figs. 12a-12c as initial topographies. (a)-(c) Color maps of steady-state landscapes. (d) Plot of histograms of junction angles extracted from the landscapes in (a)-(c).**

Below we have detailed all of the changes we have promised to make and where those changes are to be made. We look forward to addressing any additional comments provided by the referees and editors. Thank you all for your time and interest in our work.

**References not included in preprint:**

Dempsey, K.A., P.K. House, and P.A. Pearthree, 1993, Detailed Surficial Geologic Map of the Sothern Piedmont of the Tortolita Mountains, Pima County, Southern Arizona, Arizona Geological Survey Open-File Report 93-14, digital document available at https://data.azgs.arizona.edu/api/v1/collections/AOFR-1552429454964-628/ofr-93-14.pdf

Schneider, and Sagan, D., 2005, Into the Cool: Energy Flow, Thermodynamics, and Life, University of Chicago Press, 378 p.

Veitzer, S.A., B.M. Troutman, and V.K. Gupta, 2003, Power-law tail probabilities of drainage areas in river basins, Physical Review E, 68, 016123, https://doi.org/10.1103/PhysRevE.68.016123

[revised manuscript text omitted]

---

## Author Response (AR3)

**Response to referee #1:**

Line-by-line comments and responses:

Referee: "The authors clarified many issues raised in the reviews in their revised manuscript. However, one central point remains unresolved: how the particular spatiotemporal settings of the Piedmont deposits are crucial for creating low branching angles. Regarding the question of structural control: Piedmonts are geologic structures. The caption of Fig. 11, for example, reads: "Junction angles tend to be lower in areas with late-Cenozoic deposits." Thus, if the authors argue that specific properties of Piedmont deposits cause narrow branching angles, this is a clear structural argument. If not, it is simply a slope/roughness effect and should also be presented as such. For reporting the pure observation that piedmonts are gently sloped surfaces with small surface roughness and thus more elongated drainage basins with smaller branching angles, a one-page technical note would be sufficient, rather than an extensive article."

Response: Thank you for this comment. We disagree with the statement "Piedmonts are geologic structures." The definition of piedmont is "a gentle slope leading from the base of mountains to a region of flat land" (Oxford Dictionaries). Piedmonts are, therefore, exclusively defined by slope, not lithology or any other aspect of geologic structure. We have modified the caption of Fig. 11 to read: "Junction angles tend to be lower in areas with late-Cenozoic alluvial piedmont deposits." In terms of our analysis of data from real landscapes, we have established a correlation between the presence/absence of late-Cenozoic alluvial piedmont deposits and junction angles. Our proposed interpretation is that late Cenozoic alluvial piedmont deposits have a low ratio of small-scale roughness to large-scale regional slope/tilt. We have provided extensive theory and numerical modeling to bolster this interpretation, but we cannot rule out other explanations. As such, we cannot state that the sole reason for the correlation is the slope/roughness effect. Please also note that the previous version of our paper stated that we are using the presence/absence of late-Cenozoic alluvial piedmont deposits as a proxy for the roughness and slope factors that we propose are the primary control on junction angles (lines 373-379): "It is important to emphasize that the presence/absence of late-Cenozoic alluvial piedmont deposits is a proxy for what we hypothesize is the primary control on junction angles: initial $S_l/S_r$. Lower initial $S_l/S_r$ values are likely associated with late-Cenozoic alluvial piedmont deposits compared to bedrock/older deposits because such landforms tend to have a relatively low microtopographic amplitude prior to incision as a result of the avulsions and topographic diffusion associated with aggradation, e.g., local variations in elevation of ~1 m over spatial scales of ~100 m, as discussed conceptually in Section 1 and documented in the example data of Section 3.1.1."

Referee: Although Casteltort did not explicitly refer to branching angles, his convergence argument for drainage basins equivalently applies to the networks therein. Moreover, Seybold et al. explicitly discussed the slope dependence, although they neglected the influence of surface roughness.

Response: Thank you for this comment. Had we simply established a correlation between mean tributary fluvial network junction angles and the ratio of small-scale roughness to the regional slope/tilt, we would agree with the referee that our paper could be shortened. But the central argument of our paper is based on the geometric model of Howard (1971), which Castelltort and Yamato (2017) did not mention nor does the referee mention in their review of our revision. In this and in many other ways that we detailed in our response to the prior round of review, we have demonstrated that our work is not a trivial extension of Castelltort and Yamato (2017). We agree with the referee that Seybold et al. discussed the slope dependence of junction angles in their first paper (Seybold et al., 2017), and we have noted in our revision that Seybold et al. (2017) established a slope control on junction angles and concluded that the slope control was smaller than the aridity control (as their abstract states: "The correlation of mean junction angle with aridity is stronger than with topographic gradient…"). Seybold et al.'s (2017) conclusion that slope is a minor control is consistent with their Figure 4b, which shows that the correlation between junction angles and slope is extremely low ($R^2$ = 0.035). This is precisely why the roughness effect on junction angles was so important to document. No paper before ours has done that.

Referee: "We all agree, that piedmonts are gently sloped surfaces with small tolographic roughess. However, as the authors show by using Landscape Evolution Modeling and analyzing channel networks of aeolian deposits on the Loess Plateau, the specific ground into which the channels are carved is irrelevant for the narrower branching angle argument. In fact, the LEM used for the analysis does not seem to depend on any surface properties and simply applies a classical stream power incision law. Consequently, ANY landscape (Fig.12) that fulfills the slope and roughness conditions displays narrower branching angles. This point must be made very clear in a revised manuscript and not simply brushed away by "Sl/Sr values of which we did not study and don't feel qualified to comment on.

Response: Thank you for this comment. There is no evidence that "ANY" landscape that fulfills the slope and roughness conditions displays narrower branching angles. Figure 12 is a model. As such, it cannot definitively state what is true in real landscapes. In the previous round of review, the referee asked us to comment on whether some volcanic landscapes exhibit the same effect that we established between landscapes with and without late-Cenozoic alluvial piedmont deposits. Volcanic landscapes are often highly porous and, as such, erosion may be dominated by subsurface flows that do not follow the surface topography (e.g., Jefferson et al. (2010), doi:10.1002/esp.1976). We agree with the referee that any landscape that is the result of a model in which erosion is dominated by surface water flows and that exhibits no structural control or other heterogeneity and that fulfills the low roughness and/or high slope conditions likely displays narrower branching angles. But as far as real landscapes are concerned, our manuscript makes clear (lines 664-671) that many other factors besides roughness/regional slope control junction angles: "…we have left out many potential mechanisms, particularly those in bedrock landforms, that may influence junction angles, including preferential erosion along vertically oriented joints (Pelletier et al., 2009), lateral tectonic advection (Hallet and Molnar, 2001), etc. We emphasize the

role of initial Sl/Sr in this study because we believe that it is the most relevant factor for understanding the spatial variations in mean junction angles in CONUS, especially the difference between incised late-Cenozoic alluvial deposits and bedrock/older deposits. However, it is far from the only control on fluvial network junction angles." What we can say confidently is what we have said, i.e., that there is a strong correlation between the presence/absence of late-Cenozoic alluvial piedmont deposits and mean tributary fluvial network junction angles averaged at the 2.5-km scale in the conterminous United States. We posited that this was primarily the result of a low ratio of roughness to regional slope/tilt in late-Cenozoic alluvial piedmont deposits compared to other landscapes. The referee appears to want us to go further and state unequivocally that initial topography is the only reason why junction angles differ between landscapes carved into late-Cenozoic alluvial piedmont deposits versus those incised into bedrock, i.e., that lithology or other factors that may differ between bedrock and alluvial piedmont deposits play no role. We don't feel confident in making that statement. More broadly, we disagree with the referee that, by comparing landscapes formed into late-Cenozoic alluvial piedmont deposits versus those incised into bedrock, we are making a purely structural argument. Bedrock landscapes and late-Cenozoic alluvial piedmont deposits differ in many ways, one of which is slope (see the OED definition of piedmont above).

Several other technical points also need to be explained more clearly so that a reader can understand the author's procedures and arguments.

Below, the authors can find more specific comments referring to particular sections of the text:

L 13: "when orientations and slopes are computed using drainage basins rather than …" This sentence is still confusing. What is a basin's orientation? How is the basin's slope defined? Is it a basin regional slope, a mean roughness slope …, etc.? The authors should be more precise in their description.

Response: Thank you for this comment. Added: "BA properties are computed by averaging the local orientation and slope computed between each pixel and its nearest neighbors using the D8 or steepest-descent algorithm using all of the pixels in the drainage basin."

L 16/17 Contrast between Piedmont deposits and bedrock: The model argument following this sentence is purely based on the roughness/regional slope arguments and thus does not include any specific properties of bedrock vs deposits. Consequently, the argumentation should be reversed. The authors identified that regional vs. roughness slope controls branching angles, and depositional piedmonts are more likely to have gentle slopes with low surface roughness than mountainous bedrock regions.

Response: Thank you for this comment. We think it is important to separate the results of the data analysis using real landscapes from our theoretical interpretation and prioritize the data analysis because it relates to real landscapes. We have not identified in any data analysis that roughness/

slope controls branching angles. We have established that there is a strong correlation (much stronger than aridity) between junction angles and the presence/absence of late-Cenozoic alluvial piedmont deposits. We disagree with the referee's statement that the ratio of the roughness to the regional slope/tilt is not one of the "specific properties of bedrock vs (piedmont) deposits". We believe that bedrock and late-Cenozoic alluvial piedmont deposits differ in many ways, including in their ratio of initial roughness to regional slope/tilt. Given all of the possible differences between late-Cenozoic alluvial piedmont deposits and other kinds of environments/substrates, we have posited, following Castelltort and Yamato (2017), that the ratio of initial roughness to the regional slope is a likely reason (but perhaps not the sole reason) for the difference in mean junction angles. We believe this is a reasonable approach.

L.86: Climate does not form channels. As far as I remember, Seybold et al.'s argument is based on the relative dominance of diffusive channel-forming processes, which, in turn, vary systematically with climate.

Response: Thank you for this comment. We did not state that climate forms channels. We stated "Seybold et al. (2017; 2018) attributed the variation between 45° and 72° primarily to climate (with lower mean junction angles in more arid regions)." This is a correct statement.

L155: How do the authors define drainage basin averaged direction and slope? The authors should clearly describe their procedure.

Response: Thank you for this comment. Clarified: "BA properties are computed by averaging the local orientation and slope computed between each pixel and its nearest neighbors using the D8 or steepest-descent algorithm using all of the pixels in the drainage basin."

L 189: The authors say that the AVB model uses a search distance upstream/downstream until an elevation change of 10m is reached. In some cases, this can be multiple Kilometers. How do the authors consider cases when there is another junction Ustream?

Response: Thank you for this comment. The algorithm does not stop arbitrarily when it encounters another junction up or downstream. It continues past any junction. Noted in the revised manuscript.

L 191: Please clarify the orientation and slope of a single pixel is? Orientation of the steepest descent in a D-8 flow Routing scheme?

Response: Thank you for this comment. Yes. D8. Noted in the revised manuscript. Please also note that the code is publicly available, so some technical questions (such as whether D8 or some other procedure is used to define orientation) can be resolved by examination of the code.